# Transitions in intensive care: Investigating critical slowing down post extubation

**Lucinda Khalil**[1]☯, **Sandip V. George** [2,3]☯ *, **Katherine L. Brown**[4], **Samiran Ray**[5], **Simon Arridge**[2]

**1** Department of Mathematics, Imperial College London, London, United Kingdom, **2** Department of Computer Science, University College London, London, United Kingdom, **3** Department of Physics, University of Aberdeen, Aberdeen, United Kingdom, **4** Cardiac Intensive Care Unit, Great Ormond Street Hospital For Children NHS Foundation Trust, London, United Kingdom, **5** Paediatric Intensive Care Unit, Great Ormond Street Hospital For Children NHS Foundation Trust, London, United Kingdom

☯ These authors contributed equally to this work.
* sandip.george@ucl.ac.uk

**Data Availability Statement:** The raw data used in this study are not publicly available due to restrictions related to data containing information that could compromise the privacy of research participants. However, interested parties may

## Abstract

Complex biological systems undergo sudden transitions in their state, which are often preceded by a critical slowing down of dynamics. This results in longer recovery times as systems approach transitions, quantified as an increase in measures such as the autocorrelation and variance. In this study, we analysed paediatric patients in intensive care for whom mechanical ventilation was discontinued through removal of the endotracheal tube (extubation). Some patients failed extubation, and required a re-intubation within 48 hours. We investigated whether critical slowing down could be observed post failed extubations, prior to re-intubation. We tested for significant increases ($p < .05$) between extubation and re-intubation, in the variance and autocorrelation, over the time series data of heart rate, respiratory rate and mean blood pressure. The autocorrelation of the heart rate showed a significantly higher proportion of increases in the group that failed extubation, compared who those who did not. It also showed a significantly higher magnitude of increase for the failed extubation group in a t-test. Moreover, incorporating these magnitudes significantly improved the fit of a logistic regression model when compared to a model that solely used the mean and standard deviation of the vital signs. While immediate clinical utility is limited, the work marks an important first step towards using dynamical systems theory to understand the dynamics of signals measured at the bedside during intensive care.

## 1 Introduction

Dynamical systems undergo a variety of state changes due to external perturbations or internal changes. Many of these are difficult to predict from the mean response of the system. Prior to certain types of state changes in closed continuous systems, called critical transitions, a phenomenon called critical slowing down (CSD) is observed [1, 2]. CSD is characterised by an increased relaxation time in the dynamics of the system, and can be measured as increases in quantifiers such as the variance and autocorrelation over time [3, 4]. These metrics which characterize this increased relaxation time due to CSD are called early warning signals (EWS).

request access to the dataset by contacting the CHIMERA hub at UCL (chimera@ucl.ac.uk). The results of the analysis for all participants are available on the github repository https://github.com/sgeorge91/ICU_CSD/.

**Funding:** This work was funded by the ESPRC for a Hub for Mathematical Sciences in Healthcare at UCL (Collaborative Healthcare Innovation via Mathematics, Engineering and AI; CHIMERA) (EP/T017791/1) awarded to SR and SA. SVG was funded by EP/T017791/1 till June 2023. LK acknowledges funding from UCL Engineering for an in2research summer placement during which part of the study was carried out. The funders had no role in study design, data collection and analysis, decision to publish, or preparation of the manuscript.

**Competing interests:** The authors have declared that no competing interests exist.

CSD has been retrospectively observed prior to transitions in ecology [5–7], astronomy [8], finance [9], neuroscience [10] and medicine [11, 12], amongst others. In the context of medicine, slowing down of recovery has been suggested as a sign of proximity to a tipping point [13]. This has been observed in contexts such as blood pressure regulation, where slow recovery from changes in blood pressure has been linked to increased risk of mortality or of ischemic stroke [13–16] and in seizure prediction, where CSD has been reported in computational, in-vitro [17, 18], as well as in real data [19]. Critical slowing down is also thought to precede transitions in psychiatric disorders, including bipolar disorder and depression [20–22].

An area where predicting transitions is of paramount importance is in healthcare, where decisions based on these predictions can have consequences on patient well-being. This is most evident in the data rich environment of critical care, where patients are monitored using multiple-modalities continuously, and where significant deterioration events are more frequent. The response of the cardio-pulmonary system and the effects of ventilatory support are often studied using the framework of nonlinear dynamical systems [23, 24], and hence events such as sudden deterioration in health or onsets of arrythmia can be studied as tipping points [12, 25]. These could possibly be anticipated due to CSD, using EWS such as increases in auto-correlation or variance [1].

About 40% of patients in the Intensive Care Unit (ICU) require mechanical ventilation at any point in time [26], with a slightly higher number reported (50%) in the UK (https://www.picanet.org.uk/annual-reporting-and-publications/annual-report-archive/). Determining the ideal point to discontinue mechanical ventilation is an important question, since both extending and prematurely withdrawing ventilatory support can lead to adverse outcomes [27]. Prolonged mechanical ventilation is associated with a large number of complications, including risk of ventilator-associated pneumonia, muscle atrophy, as well as other physiological and psychological complications [28], while premature extubation is associated with failure as described below [29].

Patient readiness for extubation is often tested at the bedside using spontaneous breathing trials, during which ventilator support is reduced or removed for a short space of time, and patient breathing effort is observed without removal of the breathing tube. However, these trials are not fail-safe, and there is a risk that the breathing tube may need to be reintroduced (re-intubation) following extubation. Reducing the need for an emergency re-intubation is important, since the procedure carries risk of adverse events, even in a critical care environment. When there is uncertainty around the success of extubation in adults, a tracheostomy is commonly performed, whereby the breathing tube is placed through the front of the neck, and the patient is given longer (periods of days to weeks) before this is reversed following rehabilitation. In paediatric populations such as those considered in this study, tracheostomies are performed infrequently because of anatomical constraints and higher risk of complications. As a result, extubation may be undertaken even in the face of clinical uncertainty. In 5–15% of cases, extubation can be unsuccessful and the breathing tube may need to be reintroduced, resulting in what is called an 'extubation failure' [30, 31]. This can be relatively immediate, or may become apparent over a period of hours. Extubation failure is associated with generally poorer outcomes including higher mortality and increased disease severity [32]. Detection of this 'extubation failure' may be useful to prepare for re-intubation, reducing the risk from an emergency procedure. Even post extubation, it is important to predict failure quickly, since a longer time to reintubation is associated with higher mortality [33].

A number of attempts have been made to predict extubation failure and re-intubation using statistical and machine learning methods. These include logistic regression, support vector machine, random forest, gradient boosting methods and neural networks. These methods showed varying degrees of success (AUC between 0.7 and 0.85) depending on the

specifics of the algorithm and the dataset that was used for prediction [34–38]. Most of these methods used a feature set consisting of measurements such as the length of ICU stay, pre-existing comorbidities, Glasgow Coma Scale score and averages of different vital signs [39].

A smaller number of studies have investigated extubation failure by studying the dynamics of vital signs time series. White et. al. demonstrated the association between lower inter-breath interval complexity and extubation failure [40]. Seely et. al. showed that the heart and respiratory rate variability recorded prior to extubation was significantly associated with extubation outcome [41]. Keim-Malpass et. al. investigated how adding variables based on the dynamics of vital signs resulted in a significant increase in the predictive capacity of a standard model based solely on static measures [42]. A number of recent studies have also attempted to predict extubation outcome using various recurrent neural networks architectures, with varying degrees of success [43–45]. It has also been shown previously that slow recovery to baseline levels of minute ventilation following successful spontaneous breathing trials is a risk factor for failure of extubation [46, 47]. Such slow recovery to baseline levels aligns with what would be observed as a result of CSD prior to critical transitions in nonlinear dynamical systems [13].

In this study, which was conducted on a sample of paediatric patients, we hypothesized that patients who failed extubations were approaching a tipping point which was preceded by CSD, which can be measured by increased autocorrelation and variance. This implies that prior to reintubation, an increased similarity between consecutive measurements and greater variability can be observed in the time series of physiological variables. Hence as patients approached re-intubation, the presence of CSD would result in their vital signs becoming more correlated and fluctuating more over time. We explored whether the presence of CSD could be established using EWS which are observed in patients in intensive care units (ICUs) post extubation and prior to re-intubation, where the measured patient responses were not masked by ventilator dynamics. Patients who did not undergo an extubation failure were also analysed for comparison, since no CSD was expected in these cases.

Prior to the study, the analysis plan was pre-registered based on analysis conducted on a single patient (ID PC0192.1). This patient failed the extubation and required re-intubation around 20 hours later. The variance and autocorrelation were calculated for three physiological variables (heart rate, respiration rate and mean blood pressure). In this case, 5 out of the 6 tests conducted produced significant results and provided evidence against the null hypothesis that CSD is not an early warning sign of critical transitions. Complete details of this investigation can be found in the pre-registration document [48]. All three variables used in the analysis have been linked to extubation outcome in the past [49–52].

## 2 Methods

### 2.1 Ethics statement

All data used were previously collected during the period from 01/01/2016 to 31/12/2018 as part of standard clinical care, in the Great Ormond Street Hospital, London. No additional data were collected for the purposes of this work, or any wider research work. All data were fully de-identified. Therefore, the need for individual patient consent was waived under GAfREC 2018 and as a result is exempt from NHS REC approval. This was approved by the institutional joint research office (Great Ormond Street Hospital ref 19HL03) and the NHS Health Research Authority (IRAS 259636 received on 24/01/2019). The dataset was accessed for the present study between 16/08/2022 and 27/02/2023.

## 2.2 Background

CSD is a phenomenon which occurs in any closed continuous model which is approaching certain types of bifurcations, such as fold bifurcations [1]. We show this in an arbitrary dynamical system approaching a fold bifurcation. Its linearized dynamics is given by

$$\frac{dx}{dt} = \lambda x + \sigma \epsilon \,, \tag{1}$$

where x is the perturbation from equilibrium, $\lambda$ is the eigenvalue, $\sigma$ is the noise amplitude and $\epsilon$ is a white noise process. This represents an Ornstein-Uhlenbeck process, the variance and autocorrelation of which are defined as

$$Var(x) = \frac{\sigma^2}{2\lambda}, ACF(\tau) = e^{-\lambda \tau} \,. \tag{2}$$

For a fold bifurcation, $\lambda \rightarrow 0$, which implies that the variance approaches $\infty$ and ACF approaches 1 [4, 53].

## 2.3 Data

The dataset consisted of ICU visits of all children admitted to the Great Ormond Street Hospital (GOSH) between 2016 and 2018. In the current work, time series of the heart rate, respiratory rate and mean blood pressure sampled every 5 seconds were used for analysis. The data for each patient also consisted of the sex, age, timestamp of extubation attempts, a timestamp for time of re-intubation, a timestamp for time of death, flags for failed extubation and death, and the ward they were admitted to [54]. The times of extubation and re-intubation were generally determined from gaps in the end tidal $CO_2$ measurements. Missing data in the end tidal $CO_2$, closest to the recorded times of extubation and re-intubation were chosen as the actual times of extubation/re-intubation. When large discrepancies were observed between the two, a clinician examined the patient records for determining a likely time of extubation, failing which the data was discarded.

The intensive care units at Great Ormond Street Hospital are divided into three distinct areas: the neonatal ICU (NICU), where premature children and those with congenital conditions are admitted; the cardiac ICU (CICU) where children with heart disease, primarily post-cardiac surgery are admitted, and the general ICU (PICU), where all other children from ages 0–18 are admitted. A flow chart describing the separation of data from the ICU into cohorts, prior to the stratification and cleaning described in the following sections, is shown in Fig 1. The characteristics of this dataset are listed in Table 1. The dataset was prepared and used for a prior data study group with the Alan Turing Institute, UK conducted in 2021. During the data study group attempts were made at predicting extubation failure using three machine learning methods, namely time series forest, random interval spectral ensemble and random forest. None of the methods used performed better than a naive baseline classifier [54]. All data were fully de-identified.

## 2.4 Pre-processing of the data

The given data set was first split into four different cohorts depending on whether the patient underwent re-intubation and/or died during a particular visit. The criteria for each cohort is as follows:

**Cohort 1**: Failed extubation *and* were re-intubated *and* did not die

**Cohort 2**: Successful extubation *and* were not re-intubated *and* did not die

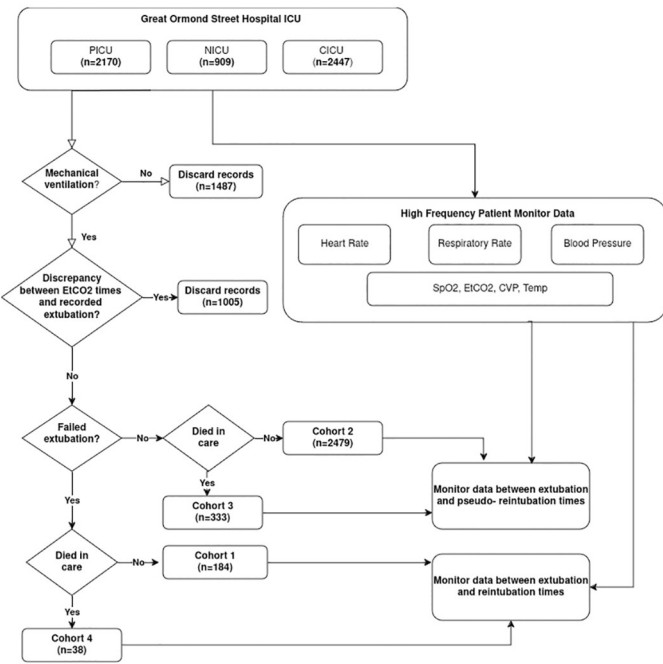

**Fig 1. Flowchart depicting the steps involved in the partitioning of patient records into cohorts.** High frequency monitor data is extracted between extubation and re-intubation times for cohorts 1 and 3 and between extubation and pseudo-reintubation times for cohorts 2 and 4. The numbers mentioned for each of the cohorts is before age based stratification and before accounting for data losses described in S1 File. SpO2: Oxygen saturation; Et CO2: End-tidal CO2; CVP: Central Venous Pressure; Temp: Temperature.

**Table 1. The characteristics of the dataset prior to pre-processing.**

|  | PICU/NICU | CICU |
|---|---|---|
| n | 3079 | 2447 |
| Male | 1757 | 1296 |
| Unplanned admission | 1783 | 558 |
| PIM%, median (IQR) | 1.52 (0.61–4.50) | 1.09 (0.78–2.39) |
| Survival | 2905 | 2379 |
| Diagnostic category n(%) |  |  |
| Cardiovascular | 128(4.16%) | 1825(74.58%) |
| Respiratory | 934(30.33%) | 277(9.28%) |
| Neurological | 552(17.9%) | 10(0.41%) |
| Gastrointestinal | 424(13.77%) | 41(1.67%) |
| Musculoskeletal | 187(6.07%) | 10(0.41%) |
| Oncology | 123(3.99%) | 18(0.74%) |
| Body wall and cavities | 125(4.06%) | 15(0.61%) |
| Other/Unknown | 606(19.68%) | 251(10.26%) |

PIM: Paediatric Index of Mortality; IQR: Interquartile range

**Cohort 3**: Extubated *and* were not re-intubated *and* died

**Cohort 4**: Extubated *and* were re-intubated *and* died after re-intubation (This group represented those in whom intensive care support was being withdrawn)

The main analysis was carried out between cohort 1 who were hypothesised to show signs of CSD and cohort 2 who were acting as a control group since it is assumed they do not reach a critical tipping point post-extubation. The same process was carried out on cohort 3 and 4 however in these cases the results are considered separately since there may be other significant biological factors affecting the dynamics of these patients. As seen in Fig 1, cohorts 1, 2, 3 and 4 have sizes of 184, 2479, 333 and 38 respectively, prior to pre-processing. For patients who did not die in care, this corresponds to a failure rate of about 6.9%.

In order to be able to make an unbiased comparison between cohort 2, the covariates of age, sex (M/F/Unknown) and ICU ward type (paediatric-PICU/neonatal-NICU/cardiac-CICU) were studied first. Fig 2 shows the distribution of both cohorts across these covariates. It is clear that the proportion of male and female instances are almost identical so this did not require stratification. Age shows more variation and since it is a key factor in determining the dynamics of the physiological data, it was stratified over. ICU ward type also shows a significant amount of variation, and therefore the testing was conducted on the entire cohorts as well the cohorts split by ICU ward type and the differences are discussed in the results section. Stratification was not carried out for cohort 3 or 4, since these cohorts were much smaller, and any stratification would have resulted in significant loss of data.

Moreover, since the calculation of the quantifiers use windows of 60 minutes of data (see section 2.5), any ICU visit where the duration between the failed extubation and critical point (either re-intubation or death) is less than 120 minutes was removed. This is so that the window size is not greater than half of the length of the time series, which would fail to pick up nuances in the data and provide inaccurate estimations [55].

In order to test the hypothesis, each instance of the control group (cohort 2) was randomly assigned a pseudo re-intubation time which was decided such that the distributions of the duration of extubated intervals are the same over cohort 1 and cohort 2. The mean and standard deviations of the measures used for analysis after pre-processing are presented in S1 File. Significant differences are seen between Cohorts 1 and 2 for all three measures, in line with previous literature [49–52].

Missing data was handled through a two step process. When the gap in observations were longer than 10 time steps, the time series was concatenated. For gaps smaller than this, the missing part was replaced with the mean of the end points of the gap. Subsequently the long term trend was identified using a Gaussian filter and subtracted from the original time series to generate the residual time series.

The number of records available for analysis was reduced significantly, since for many records the time difference between extubation and re-intubation was less than 120 minutes. More records were lost after preprocessing and during the data analysis. Different numbers of patient records were available for the different physiological variables considered. The number of records lost and the causes for the same are detailed in S1 File.

Subsequently, the time series were detrended by applying Gaussian smoothing and removing the smoothed data from the original data. Such detrending is shown to reduce false positives in detecting early warning signals from data [56]. Outliers outside $\mu \pm 3\sigma$ were discarded and replaced with $\mu$, where $\mu$ is the mean and $\sigma$ is the standard deviation.

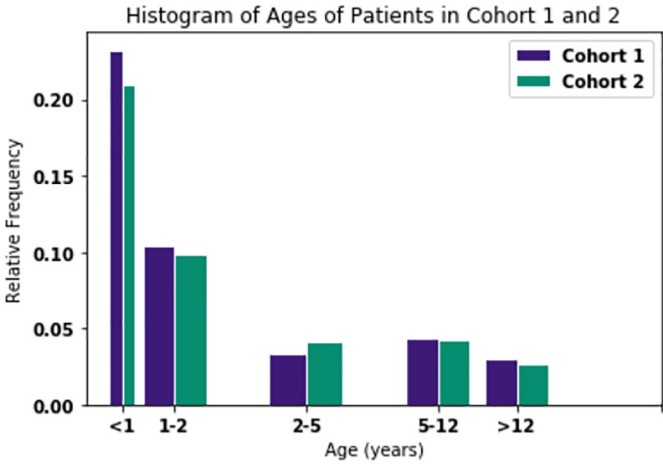

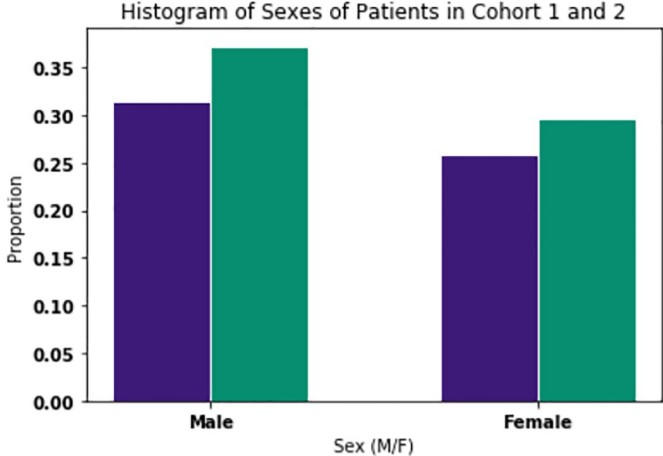

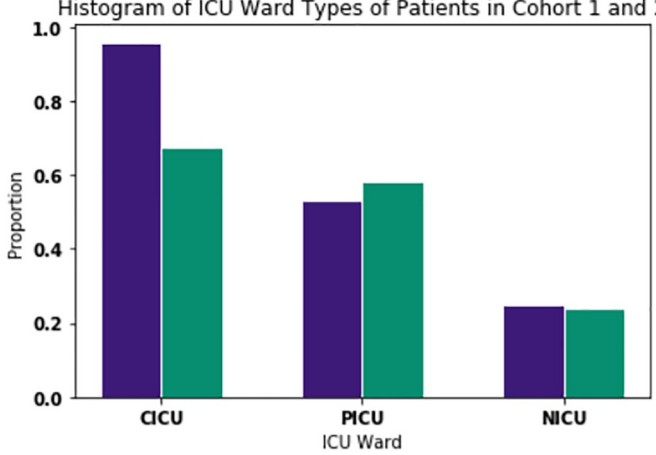

**Fig 2. Comparison of age, sex and ICU ward type of cohort 1 and cohort 2.**

## 2.5 Hypotheses

This analysis aims to investigate whether CSD can be observed post-extubation and prior to a patient being re-intubated. For the reasons discussed in Section 2.1, the following quantifiers will be studied:

1. Variance, $\sigma^2$—The presence of a period of CSD would cause an increase in variance as time approached the critical transition point. This would be due to a reduced resilience to external perturbations [57]. For a patient who did not require re-intubation, a significant increase in the variance is not expected (Note that there are instances of patients who experience a critical transition in their health but in the end did not require re-intubation, see Section 4 for further details.).

2. Autocorrelation, $r_1$—The presence of a period of CSD would also cause an increase in the lag-1 autocorrelation of the time series data. For a patient who did not require re-intubation, a significant increase in the autocorrelation is not expected.

Formally, the null and alternative hypotheses can be expressed as follows:

$H_0$: There is no period of CSD leading up to a critical tipping point

$H_1$: There is a period of CSD leading up to a critical tipping point

## 2.6 Statistical model

The method consisted of analysing the variance and autocorrelation of the heart rate (beats per minute/BPM), respiration rate (breaths per minute/BrPM) and mean blood pressure (millimeters of Mercury/mmHg).

For each of the heart rate, respiration rate and blood pressure time series, the variance and autocorrelation were calculated on sequential windows of data, slid forward by 1 minute. For a time series $X = \{x_1, x_2, \ldots, x_n\}$, a window of size m and starting time t, would be given as $X_{\{t,m\}} = \{x_t, x_{t+1}, \ldots, x_{t+m}\}$. The variance and autocorrelation of each window would then be given by

$$\sigma^{2,t} = \frac{\sum_{i=l}^{t+m} (x_i - \bar{x})^2}{(m-1)} \tag{3}$$

$$r_1^t = \frac{\sum_{i=t}^{t+m-1}(x_i - \bar{x})(x_{i+1} - \bar{x})}{\sum_{i=t}^{t+m} (x_i - \bar{x})^2} \tag{4}$$

The resulting $\sigma^{2,t}$ and $r_1^t$ form a secondary time series, as the start point of the window, t, is slid forward, which shows how these measures change over time. There is a trade-off for the choice of window size in this study, since a window that is too large will result in significant loss in data and a window that is too small will be dominated by short-term fluctuations in the time series [58, 59]. As a trade-off, primary analysis was conducted using a window size of 60 minutes and variations observed with smaller window sizes of 30 and 15 minutes were considered in an exploratory analysis and reported in S3 File.

For each of these, the modified Mann-Kendall [60] test was then carried out which calculates the Kendall-Tau correlation coefficient while also applying a correction to correct for the dependence between overlapping windows [61, 62]. This resulted in a correlation coefficient and test statistic for both the variance and autocorrelation of each physiological measure (6 combinations in total). A Holm-Bonferroni hypothesis test was conducted to correct for

**Table 2. Final number of files used for analysis each variable, cohort and ward type.**

|  | Heart Rate (BPM) | | | | Respiration Rate (BrPM) | | | | Mean Blood Pressure (mmHg) | | | |
|---|---|---|---|---|---|---|---|---|---|---|---|---|
|  | C1 | C2 | C3 | C4 | C1 | C2 | C3 | C4 | C1 | C2 | C3 | C4 |
| PICU | 47 | 354 | 32 | 12 | 37 | 305 | 26 | 12 | 21 | 164 | 22 | 7 |
| CICU | 40 | 936 | 28 | 4 | 33 | 771 | 21 | 4 | 34 | 846 | 20 | 4 |
| NICU | 14 | 185 | 13 | 2 | 11 | 164 | 9 | 1 | 4 | 23 | 3 | 1 |
| Whole | 101 | 1475 | 73 | 18 | 81 | 1240 | 56 | 17 | 59 | 1033 | 45 | 12 |

C1, C2, C3 and C4 correspond to cohorts 1, 2, 3 and 4.

multiple tests at a significance of $\alpha = 0.05$. Since the application of CSD to detect the need for reintubation is a novel approach, it is important to minimize the risk of missing clinically relevant changes. Hence the most commonly used $\alpha$ level of.05 was chosen over stricter threshold, while also correcting for multiple testing using the Holm-Bonferroni approach.

The proportions of instances which gave significant results was then calculated for each cohort, and a z-test was conducted to check whether these proportions were significantly different between the cohorts. The predictive capacity of the model was calculated using the performance metrics, positive predictive value (PPV), negative predictive value (NPV), sensitivity and specificity.

Welch's t-tests were also conducted on the corresponding distributions of Kendall-$\tau$ correlation coefficients for cohort 1 and 2 in order to explore the likelihood of these samples have the same mean.

Both the z test and the t-test were conducted to establish if Kendall-$\tau$ correlation coefficients could be used to distinguish the cohorts from each other. These tests were conducted on the results of the cohorts as a whole, as well as separating them by ICU ward type (PICU/CICU/NICU). All tests were conducted on Python 3.5, using the packages numpy, scipy and pymannkendall [60, 63, 64].

## 3 Results

In the following sections, we present the results of the statistical analyses conducted, including z-tests for significant increases in EWS, t-tests for comparing the magnitudes of increase, and a series of exploratory analyses to further investigate the trends in the data. The final number of files analyzed per ICU ward, cohort, and variable is shown in Table 2. This results in failure rates ranging from 5.4% to 6.4%, depending on the variable used without subdivision into wards, and from 3.9% to 14.8% when considering wards separately.

### 3.1 Proportions of significant results

After conducting the Mann-Kendall hypothesis test as outlined in Section 2.5, the proportion of significant results (instances which showed an increase in variance or autocorrelation over time) for each measure were calculated and compared between cohort 1 and the control group, cohort 2. Fig 3 and Table 3 show these results and how they are split among the three ICU ward types.

When the entire cohorts are considered as a whole rather than split by ICU ward type, cohort 1 has a statistically significant higher proportion of instances showing an increase in autocorrelation of the heart rate. This provides evidence to reject the null hypothesis and suggests that CSD is observed before a critical tipping point (which is taken to be the point of re-intubation) for this measure. In no other measure was there sufficient evidence to reject the

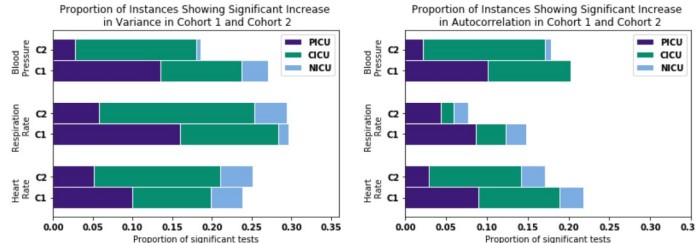

**Fig 3. Proportion of instances which showed significant increases in variance and autocorrelation respectively over the extubated period after conducting the Mann-Kendall hypothesis tests using the Holm-Bonferroni method.** Each measure of each cohort is also split by ICU ward type.

null hypothesis at the $\alpha$ level considered. However, in 5 of 6 cases, a higher proportion of trends were observed in cohort 1 than in cohort 2.

## 3.2 Distributions of correlation coefficients

The Kendall-$\tau$ correlation coefficients of each instance signal the strength of trends in variance and autocorrelation over time. Hence, a greater positive value of $\tau$ suggests a stronger presence of CSD. Fig 4 shows the distributions of these correlation coefficients for each combination of measures. We see that these distributions are roughly Gaussian, especially for cohort 2 where sample sizes are around 1000, and much larger than those of cohorts 1 which vary between 59 and 101.

For each of these sets of $\tau$ values, a Welch's t-test was conducted between cohort 1 and cohort 2 to determine whether the means of their distributions were significantly different, i.e, assuming that cohort 1 and 2 follow distributions of $N(\mu_1, \sigma_1^2)$ and $N(\mu_2, \sigma_2^2)$ respectively, the following hypotheses were tested:

$$\mathbf{H_0} : \mu_1 = \mu_2$$

$$\mathbf{H_1} : \mu_1 > \mu_2$$

The means, standard errors and p-values corresponding to each of these tests are shown in Table 4 with the significant results highlighted. This analyses were conducted on the entire cohorts as well as after splitting by ICU ward type. We see that only 1 out of 6 tests on the whole cohorts show significant results, and 3 of 18 when split over the various ICU ward types.

**Table 3. The proportions of significant Mann-Kendall hypothesis tests conducted on cohorts 1 and 2.**

| | Variance, $\sigma^2$ | | | | | | Autocorrelation, $r_1$ | | | | | |
|---|---|---|---|---|---|---|---|---|---|---|---|---|
| | HR | | RR | | ABP | | HR | | RR | | ABP | |
| | C1 | C2 | C1 | C2 | C1 | C2 | C1 | C2 | C1 | C2 | C1 | C2 |
| PICU | 0.213 | 0.221 | 0.351 | 0.236 | **0.381** | **0.170** | 0.191 | 0.119 | 0.189 | 0.180 | **0.286** | **0.134** |
| CICU | 0.219 | 0.251 | 0.303 | 0.315 | 0.176 | 0.187 | 0.250 | 0.179 | 0.091 | 0.108 | 0.176 | 0.183 |
| NICU | 0.357 | 0.330 | 0.091 | 0.304 | 0.500 | 0.261 | 0.357 | 0.232 | 0.182 | 0.110 | 0.000 | 0.305 |
| Whole | 0.238 | 0.254 | 0.296 | 0.294 | 0.271 | 0.186 | **0.238** | **0.171** | 0.148 | 0.126 | 0.203 | 0.178 |

The first 3 rows show these proportions if only one ICU ward is considered at a time. The pairs of proportions which are significantly different between cohort 1 and 2 are highlighted.

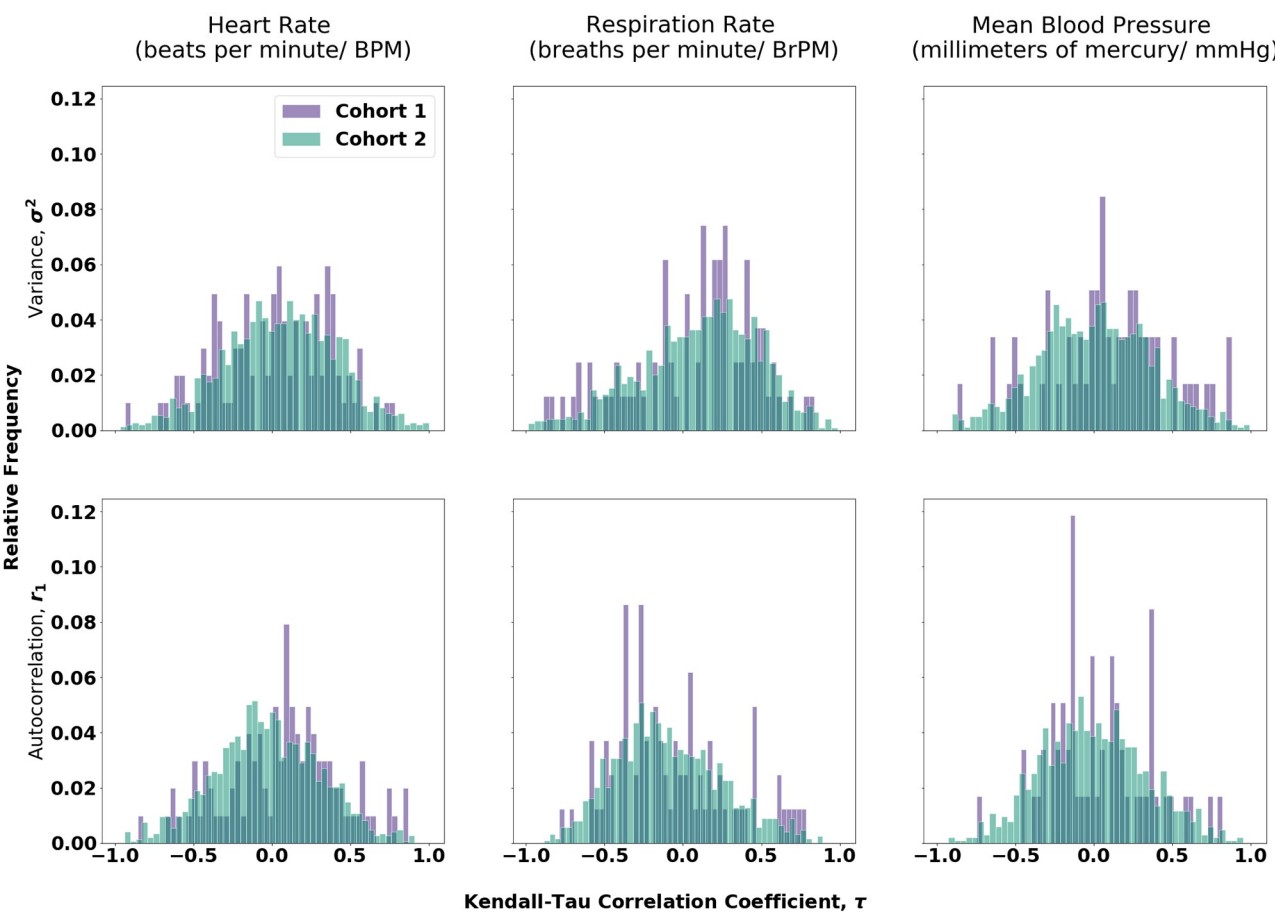

**Fig 4. Histograms showing distributions of the Kendall-Tau correlation coefficients of each time series, grouped by physiological measure (heart rate, respiration rate, mean blood pressure) and quantifier (variance, autocorrelation).** This is a comparison of the correlation coefficients of cohort 1 and 2.

The only significant test was again associated with the autocorrelation of the heart rate. This was the only measure that had a significant result in the z-test conducted in Section 3.1. Therefore, within this dataset, we see that the autocorrelation of the heart rate is the best indicator of CSD.

Overall, these results do not provide evidence to suggest that CSD may be observed in patients who are approaching a transition in their physiology, except in the autocorrelation of the heart rate.

### 3.3 Exploratory analyses

In this section, we describe the exploratory analyses that were not pre-registered as part of our main analyses. We list the broad results of our exploratory analyses here, and the details are given in supplementary material.

The first of these was to analyse the results of cohort 3 with cohort 2. Cohort 3 consisted of individuals who were extubated and subsequently died in care. The analysis was conducted for the whole time series from the point of extubation until death. None of the 6 measures showed significant differences in the proportion of significant trends between the two cohorts when they were considered as a whole, as well as when subdivided according to ICU ward.

**Table 4. Means and standard errors of Kendall-Tau correlation coefficients of cohorts 1 and 2, as well as t-statistics and p-values when Welch's T-Test is conducted.**

| | Variance, $\sigma^2$ | | | | | | | | |
|---|---|---|---|---|---|---|---|---|---|
| | Heart Rate (BPM) | | | Respiration Rate (BrPM) | | | Mean Blood Pressure (mmHg) | | |
| | $\mu_1$ (SE) | $\mu_2$ (SE) | p | $\mu_1$ (SE) | $\mu_2$ (SE) | p | $\mu_1$ (SE) | $\mu_2$ (SE) | p |
| PICU | 0.035 (0.045) | 0.026 (0.019) | 0.853 | 0.016 (0.076) | 0.044 (0.022) | 0.727 | **0.200 (0.071)** | **-0.042 (0.030)** | **0.004** |
| CICU | -0.049 (0.063) | 0.050 (0.011) | 0.125 | 0.110 (0.012) | 0.126 (0.057) | 0.819 | 0.007 (0.068) | 0.012 (0.012) | 0.939 |
| NICU | 0.130 (0.096) | 0.118 (0.028) | 0.911 | 0.139 (0.061) | 0.123 (0.030) | 0.888 | 0.207 (0.214) | 0.091 (0.069) | 0.638 |
| Whole | 0.015 (0.035) | 0.053 (0.009) | 0.298 | 0.071 (0.045) | 0.105 (0.011) | 0.462 | 0.089 (0.050) | 0.006 (0.011) | 0.104 |
| | Autocorrelation, $r_1$ | | | | | | | | |
| | Heart Rate (BPM) | | | Respiration Rate (BrPM) | | | Mean Blood Pressure (mmHg) | | |
| | $\mu_1$ (SE) | $\mu_2$ (SE) | p | $\mu_1$ (SE) | $\mu_2$ (SE) | p | $\mu_1$ (SE) | $\mu_2$ (SE) | p |
| PICU | **0.062 (0.051)** | **-0.071 (0.018)** | **0.016** | 0.015 (0.060) | -0.043 (0.021) | 0.370 | 0.079 (0.064) | -0.058 (0.027) | 0.057 |
| CICU | 0.039 (0.065) | -0.022 (0.011) | 0.360 | -0.184 (0.057) | -0.115 (0.012) | 0.248 | 0.017 (0.055) | 0.008 (0.011) | 0.872 |
| NICU | **0.175 (0.062)** | **0.027 (0.026)** | **0.041** | 0.029 (0.128) | -0.077 (0.026) | 0.310 | -0.107 (0.217) | 0.013 (0.090) | 0.634 |
| Whole | **0.069 (0.036)** | **-0.027 (0.009)** | **0.011** | -0.064 (0.041) | -0.092 (0.010) | 0.506 | 0.031(0.041) | -0.002 (0.010) | 0.438 |

The table also shows values when grouped by ICU ward type. Statistically significant results are in bold.

A similar analysis was also conducted with Cohort 4, which consisted of a very small number of individuals who were re-intubated and still died in care. In this case, 1 out of 6 variables showed significant results, namely the autocorrelations of the mean blood pressures. However, due to the small size of the cohort, these results should be viewed with caution.

We also repeated the analysis described in Section 2 with smaller window sizes of 15 and 30 minutes. We see an increase in the number of false positives in almost all quantifiers. While no significant results were seen for the 30 minute windows, the proportion of the autocorrelation of the heart rate for cohort 1 was significantly higher than for cohort 2, when using 15 minute windows. This higher proportion was also observed when data from the CICU alone was considered.

Finally, we also explored the oxygen saturation ($SpO_2$) time series for significant increases in autocorrelation and variance using hour long windows, for cohorts 1 and 2. While we see no significant differences between the two cohorts when taken together or split by ICU type, we do see that a Welch t-test showed significant differences between the distribution of correlation coefficients between the two groups, for both autocorrelation and variance of the $SpO_2$. This difference also persisted when the PICU cases were considered alone. The autocorrelation of $SpO_2$ was also significantly different between the two groups when the CICU cases were considered alone.

## 3.4 Model performance

The confusion values and various performance metrics of the measured EWS are shown in Table 5. The metrics used to evaluate the predictive capabilities of the model are Positive Predictive Value (PPV), Negative Predictive Value (NPV), Sensitivity, Specificity, Balanced Accurancy ($Acc_B$) and Area Under the Receiver Operator Characteristics (ROC) Curve (AUC). All quantifiers have values that vary between 0 and 1, with 1 indicating ideal classification performance. The PPV, NPV, Sensitivity and Specificity are affected by class imbalance, whereas $Acc_B$ and AUC are robust to this. Values close to 0.5 for the latter indicate classification no better than chance.

**Table 5. Confusion values, PPV, NPV, sensitivity and specificity of the model for cohorts 1 and 2.**

| COHORT 1 | Measure | TP | FP | FN | TN | PPV | NPV | Sensitivity | Specificity | $Acc_B$ | AUC |
|---|---|---|---|---|---|---|---|---|---|---|---|
| Variance, $\sigma^2$ | HR | 24 | 374 | 77 | 1101 | 0.060 | 0.935 | 0.238 | 0.746 | 0.492 | 0.486 |
| | RR | 24 | 365 | 57 | 875 | 0.062 | 0.939 | 0.296 | 0.706 | 0.501 | 0.499 |
| | ABP | 16 | 192 | 43 | 841 | 0.080 | 0.951 | 0.271 | 0.814 | 0.543 | 0.578 |
| Autocorrelation, $r_1$ | HR | 24 | 253 | 77 | 1222 | 0.087 | 0.941 | 0.238 | 0.828 | 0.533 | 0.587 |
| | RR | 12 | 156 | 69 | 1084 | 0.071 | 0.940 | 0.148 | 0.874 | 0.511 | 0.517 |
| | ABP | 12 | 184 | 47 | 849 | 0.061 | 0.947 | 0.204 | 0.822 | 0.513 | 0.498 |

HR: heart rate (beats per minute/BPM), RR: respiration rate (breaths per minute/BrPM), ABP: average blood flow (millimeters of Mercury/mmHg), TP: true positive, FP: false positive, FN: false negative, TN: true negative, PPV: positive predictive value, NPV: negative predictive value, $Acc_B$: balanced accuracy, AUC: Area under the ROC curve

In this study, since we are interested in only predictions due to positive trends, the ROC curve was generated by varying the threshold of the Kendall $\tau$ coefficient, while setting negative $\tau$ to a minimum negative value (effectively excluding them).

The values close to 0.5 of $Acc_B$ and AUC indicate that the model has limited predictive power. The low sensitivity values suggests that increasing variance and autocorrelation are not effective signals for predicting the need for re-intubation for patients in ICUs. The low values of PPV and high values for NPV in the study partially arise from the differences in sizes between cohorts 1 and 2. Since a significant proportion of the much larger Cohort 2 is incorrectly flagged as a positive, the chances of a positively flagged test actually being a true positive are incredibly small, as seen in the PPV.

Finally, we also show the results of logistic regression models when using multiple variables as predictors. In these models, the extubation outcome is the dependent variable, and the values of the Kendall correlation coefficients are used as the independent variables. A weight factor of 10:1 was taken to consider the relative prevalence of successful extubations relative to extubation failures. An additional model incorporating the mean values of the heart rate, respiratory rate, and blood pressure was also used for comparison. The logistic regression models used in the analysis are defined as follows.

1. $\text{ExtOutcome} \sim HR^{\tau}_{r_1} + HR^{\tau}_{\sigma^2}$

2. $\text{ExtOutcome} \sim RR^{\tau}_{r_1} + RR^{\tau}_{\sigma^2}$

3. $\text{ExtOutcome} \sim ABP^{\tau}_{r_1} + ABP^{\tau}_{\sigma^2}$

4. $\text{ExtOutcome} \sim HR^{\tau}_{r_1} + HR^{\tau}_{\sigma^2} + RR^{\tau}_{r_1} + RR^{\tau}_{\sigma^2} + ABP^{\tau}_{r_1} + ABP^{\tau}_{\sigma^2}$

5. $\text{ExtOutcome} \sim HR_{mean} + RR_{mean} + ABP_{mean}$

6. $\text{ExtOutcome} \sim \text{All Variables}$

Here $Variable^{\tau}_{r_1}$ and $Variable^{\tau}_{\sigma^2}$ represents the magnitude of the Kendall correlation coefficient for the autocorrelation and variance for each variable (HR, RR and ABP). The full results of the logistic regression models, including the logs of the odds (estimate), standard errors, z-values, p-values, degrees of freedom (N), Akaike information criterion (AIC) and $R^2$ values are summarized in Table 6. The model incorporating all variables showed a higher $R^2$ and lower AIC as compared to each of the individual models. An ANOVA showed that model 6 significantly differed from model 4 and 5 (model 4 and 6: Deviance: 162.64 p-value:<0.001; model 5 and 6: Deviance: 29.894; p-value<0.001). Since model 4 that combines the HR, RR and ABP

**Table 6. Logistic regression models using the Kendall -$\tau$ coefficients as predictors.**

| Predictor | Estimate | SE | z-value | p-value | N | AIC | $R^2$ |
|---|---|---|---|---|---|---|---|
| | | | 1. ExtOutcome $\sim HR^{\tau}_{r_1} + HR^{\tau}_{\sigma^2}$ | | | | |
| intercept | **-0.346** | **0.041** | **-8.407** | **<0.001** | | | |
| $HR^{\tau}_{r_1}$ | **0.906** | **0.125** | **7.221** | **<0.001** | | | |
| $HR^{\tau}_{\sigma^2}$ | **-0.511** | **0.121** | **-4.217** | **<0.001** | 1579 | 3357.2 | .042 |
| | | | 2. ExtOutcome $\sim RR^{\tau}_{r_1} + RR^{\tau}_{\sigma^2}$ | | | | |
| intercept | **-0.373** | **0.047** | **-8.009** | **<0.001** | | | |
| $RR^{\tau}_{r_1}$ | 0.174 | 0.133 | 1.303 | 0.192 | | | |
| $RR^{\tau}_{\sigma^2}$ | -0.160 | 0.121 | -1.320 | 0.187 | 1323 | 2789.1 | 0.004 |
| | | | 3. ExtOutcome $\sim ABP^{\tau}_{r_1} + ABP^{\tau}_{\sigma^2}$ | | | | |
| intercept | **-0.577** | **0.052** | **-10.998** | **<0.001** | | | |
| $ABP^{\tau}_{r_1}$ | -0.090 | 0.179 | -0.500 | 0.617 | | | |
| $ABP^{\tau}_{\sigma^2}$ | **0.701** | **0.162** | **4.315** | **<0.001** | 1093 | 2131.6 | 0.023 |
| | | | 4. ExtOutcome $\sim HR^{\tau}_{r_1} + HR^{\tau}_{\sigma^2} + RR^{\tau}_{r_1} + RR^{\tau}_{\sigma^2} + ABP^{\tau}_{r_1} + ABP^{\tau}_{\sigma^2}$ | | | | |
| intercept | 0.140 | 0.074 | 1.874 | 0.061 | | | |
| $HR^{\tau}_{r_1}$ | **0.695** | **0.219** | **3.169** | **0.001** | | | |
| $HR^{\tau}_{\sigma^2}$ | -0.303 | 0.226 | -1.341 | 0.180 | | | |
| $RR^{\tau}_{r_1}$ | 0.003 | 0.217 | 0.014 | 0.989 | | | |
| $RR^{\tau}_{\sigma^2}$ | -0.317 | 0.185 | -1.714 | 0.086 | | | |
| $ABP^{\tau}_{r_1}$ | 0.058 | 0.256 | 0.227 | 0.820 | | | |
| $ABP^{\tau}_{\sigma^2}$ | **0.730** | **0.229** | **3.189** | **0.001** | 454 | 1196.3 | .066 |
| | | | 5. ExtOutcome $\sim HR_{mean} + RR_{mean} + ABP_{mean}$ | | | | |
| intercept | **-7.968** | **0.722** | **-11.042** | **<0.001** | | | |
| $HR_{mean}$ | **0.021** | **0.004** | **5.466** | **<0.001** | | | |
| $RR_{mean}$ | **0.039** | **0.009** | **4.549** | **<0.001** | | | |
| $ABP_{mean}$ | **0.056** | **0.006** | **9.125** | **<0.001** | 454 | 1057.6 | 0.322 |
| | | | 6. ExtOutcome $\sim$ All Variables | | | | |
| intercept | **-8.287** | **0.755** | **-10.969** | **<0.001** | | | |
| $HR^{\tau}_{r_1}$ | **0.627** | **0.246** | **2.553** | **0.011** | | | |
| $HR^{\tau}_{\sigma^2}$ | -0.118 | 0.250 | -0.470 | 0.638 | | | |
| $RR^{\tau}_{r_1}$ | -0.102 | 0.243 | -0.421 | 0.674 | | | |
| $RR^{\tau}_{\sigma^2}$ | **-0.491** | **0.209** | **-2.339** | **0.019** | | | |
| $ABP^{\tau}_{r_1}$ | -0.319 | 0.283 | -1.128 | 0.259 | | | |
| $ABP^{\tau}_{\sigma^2}$ | **1.085** | **0.263** | **4.129** | **<0.001** | | | |
| $HR_{mean}$ | **0.026** | **0.004** | **6.018** | **<0.001** | | | |
| $RR_{mean}$ | **0.039** | **0.009** | **4.305** | **<0.001** | | | |
| $ABP_{mean}$ | **0.053** | **0.006** | **8.269** | **<0.001** | 454 | 1039.7 | 0.370 |

When a predictor was significant in the model ($p < .05$), the corresponding row is shown in bold. The Nagelkerke $R^2$ was used to estimate the goodness of fit.

N: No. of samples; AIC: Akaike information criterion; ExtOutcome: Extubation Outcome.

variables performed better than models 1,2 and 3 individually, a one-sided t-test was conducted to check if Cohort 1 had a higher strength of positive transitions across variables (defined as the sum of positive $\tau$ for all variables) as compared to Cohort 2, yielding significant results (t-statistic = 2.056, p-value = 0.022). Moreover, the AUC of the combined strength of

positive transitions was 0.597, higher than the AUC for the individual variables. The logistic regression was conducted using the glm function in R version 4.3.1.

## 4 Discussion

### 4.1 Summary and significance

In this paper, we explored whether paediatric patients who fail extubation in the ICU are approaching a critical transition in their physiology. We studied this by checking for signs of CSD in the dynamics of their heart rate, respiratory rate, and blood pressure. We tested these variables for increases in variance and autocorrelation over time. The proportion of significant increases was calculated and compared to the corresponding proportion in a stratified sample of individuals who did not fail extubation. We found that 1 out of the 6 tests, namely the autocorrelation of the heart rate showed a significantly larger proportion of positive tests in the individuals who failed extubation. The autocorrelation of the heart rate also showed a significantly higher magnitude of increase over all patients in Cohort 1 over time, as compared to Cohort 2. When calculating the model performance using metrics such as sensitivity and specificity, we observe low sensitivities and positive predictive values, suggesting limited clinical utility in using these metrics by themselves for predicting an approaching transition. Similar analyses in Cohort 3 yielded worse results with none of the proportion of trends in Cohort 3 being significantly larger than Cohort 2. While Cohort 4 showed 1 out of 6 results to have significantly larger proportions than Cohort 2, the small sample sizes make these results more suspect. In our exploratory analysis, we observe that when a smaller window size of 15 minutes was used, the autocorrelation of the heart rate showed significantly larger proportion of positive tests in the individuals who failed extubation. This effect, however, was not observed when 30 minute windows were used instead. We also observed that the $SpO_2$ time series showed a significantly higher value for the magnitude of increase in auto-correlation and variance in individuals who failed extubation, as compared to those who did not. Finally, we also observe in a logistic regression model that using the values of correlation coefficients leads to a significantly better model than one that uses only the mean values of the measures.

One of the interesting features of the study is the distributions of the Kendall $\tau$ coefficients in Fig 4. Apart from strong false positives and negatives, we also see a number of negative trends in both cohorts 1 and 2. There is some evidence that such negative trends could occur in autocorrelation in systems exhibiting flickering prior to transitions or in both autocorrelation and variance as a consequence of processes that exhibit critical speeding up [65–67]. The variance is also known to decrease when there is insufficient data and the fluctuations in the system are dominated by low frequencies or when the data is highly noisy [68]. In complex biological data, such as this both issues related to data, as well as competing processes with opposite effects could lead to such negative trends.

While our analysis detected some statistically significant differences, notably in the autocorrelation of heart rate, low values for performance metrics indicate limited utility for prediction. Taken together, our findings indicate that CSD based metrics alone are not sufficient to predict extubation outcome and may require either substantial methodological refinement or combining with other complimentary metrics to yield acceptable predictability.

Our exploration is an important step in various directions. To the best of our knowledge, it is the first study that explores CSD in the dynamics of critically ill patients. It also adds to the growing literature on using CSD and EWS to study transitions in medicine [12, 19, 69]. Moreover, it is among the first studies that use dynamical systems theory to study problems in critical care. It also identifies variables and quantifiers that could aid in enhancing the prediction of extubation outcome in the future. It uses a data agnostic approach, with the understanding

that the presence of CSD can offer information to add to the expert knowledge of a clinician making a decision. The significance of these findings extends to addressing challenges posed by public health crises such as the COVID-19 pandemic, where the demand for ventilators exceeded supply.

## 4.2 Relation to existing literature

While our work uses a novel technique different from any previous explorations on critical care data, our observations conform with the results in the literature. For instance, in neonates, Goel et. al. observed that the heart rate characteristics index, a score that incorporates the variability, asymmetry, and entropy of ECG data, was correlated to extubation outcome and enhanced predictability in regression models [51] Miu et. al. observed that for the first 60 minutes after reintubation, the $SpO_2$ significantly differed between patients requiring reintubation and those who did not. They also observed significantly different mean heart rate, $SpO_2$, and breathing frequency between the extubation failure and success group leading up to extubation [35].

## 4.3 Reasons for low model performance

While the results do not conclusively detect CSD in critically ill patients, they do show an effect in the expected direction in almost all cases. However, sensitivity is generally low, with even the best performing indicators detecting less than a third of extubation failures, implying that a significant number of extubation failures could go undetected. In the context of critical care, missed detections are more crucial than false detections, since the former can prevent timely interventions. Hence it may be more important for future studies to improve sensitivity, even at the cost of specificity.

A major reason for the observed low model performance could be that the timing of re-intubation is subjective. Hence, clinicians may be conservative in estimating when a person should be re-intubated and could lead to re-intubation before signs of CSD are strong enough to detect. Consequently, even if patients were approaching a critical transition point, the strength of EWS they would display and the time at which they would appear, would vary considerably. Moreover, this study assumes that the time scale at which the dynamics evolves for all three signals considered are the same, an assumption which is not true.

Studies that examine CSD in medical data are limited by the definition of what constitutes a positive. In many cases, as mentioned above, the decision to re-intubate may have been taken before signs of CSD become prominent. These would be treated as false negatives in our study, but may not be so in reality. In other cases, CSD may have happened in reality, since the patient was approaching a transition point, but the system could have steadied itself due to internal regulatory mechanisms of the body. Moreover, a number of potential confounding factors could influence the dynamics of the vital signs considered in this work. These include influences from drug based interventions which could alter heart rate and blood pressure, as well as the patient's fluid balance which relates to respiratory and cardiovascular stability [70, 71]. These cases would not be false positives in the correct sense, but would be considered as such in our study [4, 72]. An example of this can be observed in the case of patient FR3426 who showed significant increases in 5 out of 6 of the measures considered. As Fig 5 shows, the patient showed a major change in physiology at around 06:00 on the 29$^{th}$, even though this did not lead to subsequent re-intubation or death. This is made clear by the steep increase in variance which begins at this time. This would be an example of a false positive in our study, which may not be so in reality.

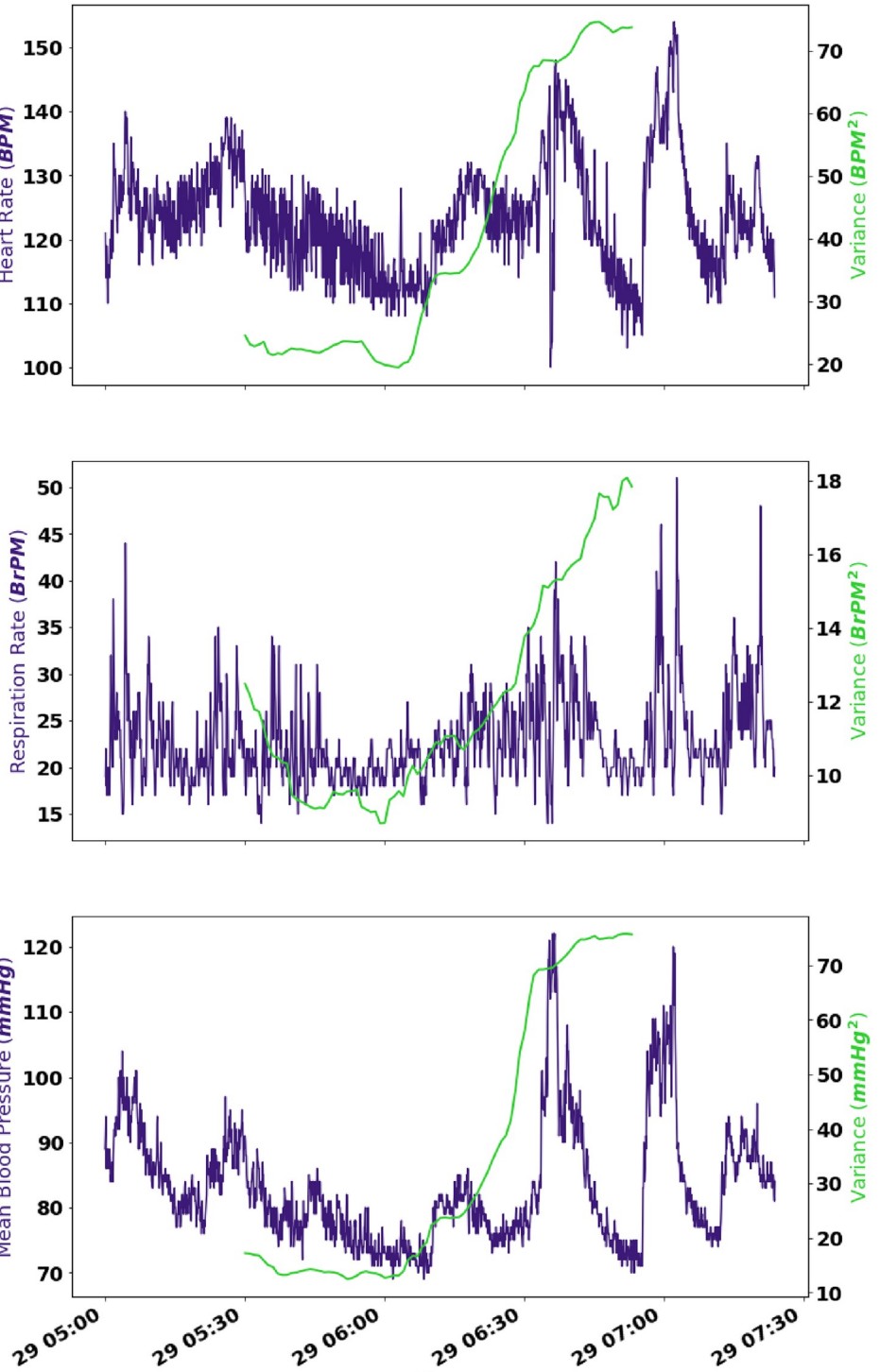

**Fig 5. Raw data (in purple) of the heart rate, respiration rate and mean blood pressure, as well as the variance (in green) calculated over a moving window, for the first visit of patient FR3426 between the time of extubation and pseudo re-intubation.**

Although increasing variance and autocorrelation have analytically been proven to be signals of CSD, these have often been difficult to observe in complex biological systems [69, 73, 74]. Several factors may contribute to this. For instance, the system itself may not be undergoing a bifurcation that is expected to be theoretically preceded by CSD [75]or the variables under consideration may not be in the direction of slowest recovery, where EWS are most prominent [76–78]. Moreover, competing processes could lead to critical speeding up of the dynamics or flickering of the transition which could mask detection of EWS in the system [65, 67, 69].

Beyond these theoretical considerations, practical issues such as under-sampling, insufficient data and time scale of occurrence could lead to false negatives in real time series [79–81]. False positives on the other hand can occur due to serial correlations and time dependent trends [56, 61]. While detrending can remove long term trends, too small a bandwidth removes important trends, and increases false negatives whereas too long a bandwidth may not remove all irrelevant trends and consequently increase false positives [82]. It has been suggested that in systems where naturally strong fluctuations exist in the system response, other early warning signals that quantify flickering could be more useful in detecting transitions [13, 66]. Metrics derived from detrended fluctuation analysis and significance testing using bootstrapping methods have both shown potential as robust approaches for detecting CSD in real-world scenarios [4, 12, 58].

In our study we attempted to overcome these difficulties by systematically examining and overcoming them in a single test case [48]. However the largely negative results of our study on the entire dataset seems to suggest that either the issues identified on the single dataset did not generalize to the whole system. To avoid this, future studies in such complex biological datasets may benefit from using a small, representative subset (perhaps 5%) of the dataset to optimize parameters for identifying CSD.

## 4.4 Limitations

While the present exploration is unique in many ways, it represents a preliminary approach towards using dynamical systems to study extubation failure. Although the high sampling frequency of the vital signs time series used in this work is rare in studies on extubation failure, the sample size of the data is comparatively small. This is particularly true in comparison to the MIMIC datasets which have been used extensively to study extubation failure [83, 84]. Furthermore, we use only data from a single paediatric hospital in London, and our results may not directly translate to an adult population.

As mentioned above, a major limitation in studies on extubation is in identifying true physiological transition points. In clinical practice, the decision to re-intubate is made preemptively in anticipation of potential instability, rather than waiting for definitive evidence of a transition. As a result transition points in this study may not indicate a change in the dynamics of the system, unlike in other fields where EWS have been detected. This study uses the timing of re-intubation as an approximate transition point, which may often be far from the true transition point.

## 4.5 Future directions

We have a few suggestions for future exploration based on the results of our study. We have largely null results for most of our exploration, except for the autocorrelation of the heart rate. While reasons for this have been discussed in the paragraphs above, one other possible avenue to explore could be analysis of waveform data from the ICU. The time series used presently are

quantities derived from waveform data like ECG or breathing waveform, and as such the non-linear dynamics could be obscured in these derived quantifiers.

To overcome the limitation of the subjectivity associated with transition points, future studies could focus on instances where emergency reintubation was required. In such cases, the actual point of reintubation may be much closer to the transition point than in a general sample. Another possibility could be to quantify markers of instability, such as deviations in physiological baselines using methods such as change point analysis. Only reintubations that also include significant quantitative change would then be checked for early warning signals, as done in studies on CSD in other clinical contexts [85].

A significant outcome of the present study is the improvement seen by using CSD variables in a simple logistic regression model. At the group level, the combination of CSD variables showed lower AIC and higher $R^2$ than models using individual CSD variables. This difference persisted even when the degrees of freedom were kept the same. This indicated that a combination of variables could perform better than individual CSD variables to detect extubations. Moreover Cohort 1 showed a higher strength of positive transitions across variables as compared to Cohort 2, and a higher predictive capacity (measured using AUC) as compared to the individual variables. This indicates that using a combination of EWS at an individual level could reduce misdetection and increase sensitivity in the sample. Setting an optimal threshold for this combined strength of positive transitions could increase sensitivity even at the cost of specificity, which may be desirable in clinical contexts. We also see that adding CSD metrics to the mean levels of the variables in the logistic regression model also showed a significant improvement over a model without them. At an individual level this could indicate that the mean and CSD metrics calculated over a sliding window could predict extubation outcome better than using either of them alone. Going even further, while CSD metrics on their own may not be a reliable predictor of extubation outcome, they could constitute novel features in machine learning based prediction models and enhance their predictive power [42]. In a real time setting, binary outcomes could be replaced with a continuous measure that gives the probability of failure, which could aid in clinical decision support [86]. In the context of the CSD, this could be the Kendall $\tau$ values, or the deviations from baseline for a sliding window mean, or prediction probabilities for a machine learning algorithm.

Another suggested path for future exploration is to study differences in patients who exhibit CSD in their dynamics from those who do not. Such differences could include medical comorbidities such as asthma, heart arrhythmia or pregnancy stressors in neonatal admissions. Beyond extubation, testing CSD prior to more objective markers where the intervention point is not arbitrary, such as cardiac arrests, is likely to be a fruitful direction for further research. While detecting CSD in extubated patients is an important first step, it is important to conduct an extensive analysis to find how far in time before reintubation CSD can be detected, since a longer time to reintubation is associated with higher mortality [33]. In fact, it would be even more clinically relevant to test whether slowing down exists leading up to extubation in patients who are likely to fail extubation in future.

## 4.6 Conclusions

Predicting extubation outcome is an important question that has been studied extensively in the past, with mixed results. The results of the present study found limited evidence to support the use of CSD based metrics to predict extubation outcome, with only the autocorrelation of the heart rate showing significant differences between the groups. These largely null results highlight the challenges of detecting critical slowing down in complex biological systems in general, and more so in critical care, where confounding factors often obscure potential early

warning signals. Future studies should be wary of these challenges and should carefully consider whether investigating CSD in clinical contexts is likely to yield meaningful benefits. Our findings point towards the need for a marked change in the way CSD based metrics are currently used for prediction in the context of medicine. For instance, combining CSD based metrics along with other indicators of extubation failure, or using them as one of many features in a machine learning algorithm could lead to improved prediction. This could perhaps be the way for future researchers looking to use CSD metrics or other measures from dynamical systems theory for prediction in the ICU.

## Supporting information

**S1 File. Characteristics of the dataset.** In this supplementary we discuss the reasons for missing records in the data and present the results of how the mean characteristics of the data vary between Cohorts 1 and 2.
(PDF)

**S2 File. Comparisons of cohorts 3 and 4 with control.** In this supplementary we explore how the proportions of records that show critical slowing down in cohorts 3 and 4 vary from cohort 2.
(PDF)

**S3 File. Variation with smaller window sizes.** In this supplementary we explore how the results vary when using smaller window sizes of 15 and 30 minutes.
(PDF)

**S4 File. Critical slowing down in oxygen saturation data.** In this supplementary we explore whether critical slowing down can be detected in oxygen saturation data, leading up to extubation failure.
(PDF)

## Acknowledgments

We acknowledge John Booth, Thalitha Grossman and Yael Feinstein for their work in helping to prepare and validate the data set used in this study. We gratefully acknowledge comments from an anonymous reviewer on this manuscript which has helped improve the discussion section considerably.

## Author Contributions

**Conceptualization:** Lucinda Khalil, Sandip V. George, Simon Arridge.

**Data curation:** Samiran Ray.

**Formal analysis:** Lucinda Khalil, Sandip V. George.

**Investigation:** Lucinda Khalil, Sandip V. George, Samiran Ray.

**Methodology:** Lucinda Khalil, Sandip V. George, Katherine L. Brown, Samiran Ray, Simon Arridge.

**Supervision:** Sandip V. George, Katherine L. Brown, Samiran Ray, Simon Arridge.

**Visualization:** Lucinda Khalil.

**Writing – original draft:** Lucinda Khalil, Sandip V. George.

**Writing – review & editing:** Sandip V. George, Katherine L. Brown, Samiran Ray, Simon
Arridge.

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
