## [Decision Letter · Decision Letter 0]

2 Jun 2024

PONE-D-24-07937Transitions in intensive care: Investigating critical slowing down post extubationPLOS ONE

Dear Dr. George,

Thank you for submitting your manuscript to PLOS ONE. After careful consideration, we feel that it has merit but does not fully meet PLOS ONE’s publication criteria as it currently stands. Therefore, we invite you to submit a revised version of the manuscript that addresses the points raised during the review process.

enhancing the method section with a graph to illustrate the step-by-step data collection process can significantly boost clarity. Moreover, further explanation of both the statistical and clinical methodologies and their correlation is needed.

Please include the following items when submitting your revised manuscript:A rebuttal letter that responds to each point raised by the academic editor and reviewer(s). You should upload this letter as a separate file labeled 'Response to Reviewers'.A marked-up copy of your manuscript that highlights changes made to the original version. You should upload this as a separate file labeled 'Revised Manuscript with Track Changes'.An unmarked version of your revised paper without tracked changes. You should upload this as a separate file labeled 'Manuscript'.If applicable, we recommend that you deposit your laboratory protocols in protocols.io to enhance the reproducibility of your results. Protocols.io assigns your protocol its own identifier (DOI) so that it can be cited independently in the future. For instructions see: https://journals.plos.org/plosone/s/submission-guidelines#loc-laboratory-protocols. Additionally, PLOS ONE offers an option for publishing peer-reviewed Lab Protocol articles, which describe protocols hosted on protocols.io. Read more information on sharing protocols at https://plos.org/protocols?utm_medium=editorial-email&utm_source=authorletters&utm_campaign=protocols.

We look forward to receiving your revised manuscript.

Kind regards,

Lalit Gupta

Academic Editor

PLOS ONE

“This work was funded by the ESPRC for a Hub for Mathematical Sciences in Healthcare at UCL (Collaborative Healthcare Innovation via Mathematics, EngineeRing and AI; CHIMERA) (EP/T017791/1) awarded to SR and SA. SVG was funded by  EP/T017791/1 till June 2023. LK acknowledges funding from UCL Engineering for an in2research summer placement during which part of the study was carried out.”

3. For studies involving third-party data, we encourage authors to share any data specific to their analyses that they can legally distribute. PLOS recognizes, however, that authors may be using third-party data they do not have the rights to share. When third-party data cannot be publicly shared, authors must provide all information necessary for interested researchers to apply to gain access to the data. (https://journals.plos.org/plosone/s/data-availability#loc-acceptable-data-access-restrictions)

Additional Editor Comments:

enhancing the method section with a graph to illustrate the step-by-step data collection process can significantly boost clarity. Moreover, further explanation of both the statistical and clinical methodologies and their correlation is needed.

Reviewers' comments:

Reviewer's Responses to Questions

**Comments to the Author**

1. Is the manuscript technically sound, and do the data support the conclusions?

Reviewer #1: Yes

Reviewer #2: Yes

Reviewer #3: Yes

2. Has the statistical analysis been performed appropriately and rigorously? 

Reviewer #1: I Don't Know

Reviewer #2: Yes

Reviewer #3: Yes

3. Have the authors made all data underlying the findings in their manuscript fully available?

Reviewer #1: No

Reviewer #2: Yes

Reviewer #3: Yes

4. Is the manuscript presented in an intelligible fashion and written in standard English?

Reviewer #1: Yes

Reviewer #2: Yes

Reviewer #3: Yes

5. Review Comments to the Author

Reviewer #1: Thanks for the interesting idea and elaborative work.

I believe your methodology needs more explanation and correlation with the clinical basis.

I believe that the statistical methodology needs more explanation in correlation with the medical part.

Reviewer #2: Dear authors, many thanks for your valuable work. Could you please, add a graph to illustrate the method section clearly? I think it will support the method section. It will focus on the process of data collection step by step.

Reviewer #3: This study is interesting for the readers.

It is scientifically sound and contains sufficient interest and originality to merit publication.

This paper is an important contribution and I recommend that it be accepted for publication.

6. PLOS authors have the option to publish the peer review history of their article (what does this mean?). If published, this will include your full peer review and any attached files.

Reviewer #1: **Yes: **fadi aljamaan

Reviewer #2: No

Reviewer #3: No

---

## [Author Response · Author response to Decision Letter 0]

28 Jun 2024

We thank the editor and the referees for their questions and comments. Modifications made in response to these have improved the manuscript considerably, and we hope that the referees and the editor are satisfied with the revisions made. Queries (Q) and responses (R) are listed below.

Editor:

Q. enhancing the method section with a graph to illustrate the step-by-step data collection process can significantly boost clarity. Moreover, further explanation of both the statistical and clinical methodologies and their correlation is needed.

R. Thank you for these comments. In line with these suggestions, a new figure (Figure 1) has been included in the revised version of the manuscript, detailing the steps of data collection and segregation into data cohorts. In addition, we have expanded the introduction by incorporating references to previous work, which motivates how the statistical techniques used in this manuscript can be used to answer questions in the context of medicine in general and critical care in particular. These are elaborated further below.

Reviewer #1: 

Q. Thanks for the interesting idea and elaborative work.

 I believe your methodology needs more explanation and correlation with the clinical basis.

 I believe that the statistical methodology needs more explanation in correlation with the medical part.

R. Thank you for this comment. In line with this suggestion, we have now expanded the introduction to motivate how the statistical techniques that have been used in this study are relevant to medicine and critical care. In particular we have added references to previous literature which suggests that extubation may be accompanied by slowing of recovery, as expected in critical slowing down. The modifications made to the introduction are mentioned below.

‘In the context of medicine, slowing down of recovery has been suggested as a sign of proximity to a tipping point [13]. This has been observed in contexts such as blood pressure regulation, where slow recovery from changes in blood pressure has been linked to increased risk of mortality or of ischemic stroke [13–16].‘

‘It has been shown previously that slow recovery to baseline levels of minute ventilation following successful spontaneous breathing trials is a risk factor for failure of extubation [30, 31]. Such slow recovery to baseline levels is in line with CSD prior to critical transitions in nonlinear dynamical systems [13].’

Reviewer #2: 

Q. Dear authors, many thanks for your valuable work. Could you please, add a graph to illustrate the method section clearly? I think it will support the method section. It will focus on the process of data collection step by step.

R. We thank the referee for their suggestion. In line with this comment, we have now added a figure (Figure 1) in the revised manuscript to illustrate the data collection and segregation into cohorts.

Reviewer #3: 

Q. This study is interesting for the readers.

 It is scientifically sound and contains sufficient interest and originality to merit publication.

 This paper is an important contribution and I recommend that it be accepted for publication.

R. We thank the referee for their kind comments.

---

## [Editor Report · Decision Letter 1]

30 Jul 2024

PONE-D-24-07937R1Transitions in intensive care: Investigating critical slowing down post extubationPLOS ONE

Dear Dr. George,

Thank you for submitting your manuscript to PLOS ONE. After careful consideration, we feel that it has merit but does not fully meet PLOS ONE’s publication criteria as it currently stands. Therefore, we invite you to submit a revised version of the manuscript that addresses the points raised during the review process.

The revised manuscript has improved significantly with the addition of the new figure and expanded introduction. However, further clarification and expansion of the methods, results and discussion sections would help strengthen the manuscript and make it suitable for publication.This is to provide more context, justification and discussion around the findings to give readers a complete picture of the study

We look forward to receiving your revised manuscript.

Kind regards,

Lalit Gupta

Academic Editor

PLOS ONE

Additional Editor Comments:

Introduction

The introduction could be further expanded to provide more clinical context and motivation for using critical slowing down (CSD) to study extubation failure. Specifically: Provide statistics on extubation failure rates and associated complications to highlight the clinical importance of the problem

Discuss in more detail how slowing of recovery dynamics has been observed in other medical contexts as a sign of proximity to a critical transition, to strengthen the rationale for investigating CSD in extubation failure

Mention any prior work that has looked at dynamics of vital signs around extubation, to position this study in the context of existing literature.

Methods

The data collection and cohort segregation process could be explained in more detail in the main text, in addition to the new Figure 1. Specifically:Provide the total number of extubations analyzed and how many failed, to give a sense of the sample size.Clarify if any exclusion criteria were applied (e.g. minimum duration of data before/after extubation).

The statistical methods section could be expanded to:

Define the CSD metrics (autocorrelation, variance) more precisely and explain how they are calculated from the vital sign time series.

Justify the choice of a 5% significance threshold for the CSD metric increases.

Describe the logistic regression model in more detail - what were the predictor variables and how were the magnitudes of CSD metric increases incorporated?

Results

Report the total number of extubations analyzed and failure rate, to provide context for the findings.

Providing the mean and standard deviation of the CSD metric increases for the success and failure groups, not just the p-values and effect sizes.

Reporting the full logistic regression model results (coefficients, p-values, odds ratios) rather than just the model comparison.

Discussion

Acknowledge the limitations of the study, such as the small sample size and single-center nature of the data.

Discuss potential confounding factors that could influence the vital sign dynamics, such as medications, fluid status, etc.

Speculate on how CSD metrics could potentially be used clinically in the future (e.g. as part of a predictive model for extubation failure) while acknowledging the current limitations.

---

## [Author Response · Author response to Decision Letter 1]

13 Sep 2024

Response letter: PONE-D-24-07937R1

We thank the editor for their detailed review of the manuscript which has helped improve it significantly. We have addressed specific concerns raised by the editor in the following paragraphs. The queries have been shown as Q, responses as R and text from the manuscript with T.

Q. The revised manuscript has improved significantly with the addition of the new figure and expanded introduction. However, further clarification and expansion of the methods, results and discussion sections would help strengthen the manuscript and make it suitable for publication.This is to provide more context, justification and discussion around the findings to give readers a complete picture of the study

R. We thank the editor for their kind comments on the revisions made. We believe that the changes made in response to the comments below have improved the manuscript considerably and makes it more suitable for publication.

Q. Introduction

Q1. The introduction could be further expanded to provide more clinical context and motivation for using critical slowing down (CSD) to study extubation failure. Specifically: Provide statistics on extubation failure rates and associated complications to highlight the clinical importance of the problem

R. We thank the editor for these comments. The introduction has been expanded considerably to include more text providing motivation for using CSD to study extubation failure, and the statistics of extubation failure and complications associated with it. Reported extubation failure rates vary between 5-15% of cases. This failure can result in poorer outcomes, including higher mortality and increased disease severity. On the other hand prolonged mechanical ventilation is associated with ventilator-associated pneumonia and muscle atrophy. Hence, detecting extubation failure quickly is critical, as delayed re-intubation correlates with higher mortality. Critical slowing down (CSD) is associated with sudden transitions in dynamical systems, and has been observed in multiple real world contexts. However, no investigation has been conducted so far to explore CSD prior to extubation failure. Further motivation for why CSD could work in this context is provided in response to Q2 and Q3.

T. Prolonged mechanical ventilation is associated with a large number of complications, including risk of ventilator-associated pneumonia, muscle atrophy, as well as other physiological and psychological complications[28], while premature extubation is associated with failure as described below[29].

In 5-15% of cases, extubation can be unsuccessful and the breathing tube may need to be reintroduced, resulting in what is called an 'extubation failure'[30,31]. This can be relatively immediate, or may become apparent over a period of hours. Extubation failure is associated with generally poorer outcomes including higher mortality and increased disease severity[32]. Detection of this ’extubation failure’ may be useful to prepare for re-intubation, reducing the risk from an emergency procedure. Even post extubation, it is important to predict failure quickly, since a longer time to reintubation is associated with higher mortality [33].

Q2. Discuss in more detail how slowing of recovery dynamics has been observed in other medical contexts as a sign of proximity to a critical transition, to strengthen the rationale for investigating CSD in extubation failure.

R. The introduction has been expanded with examples of how slowing down of recovery has been observed in other medical contexts, such as blood pressure regulation, seizure prediction and in psychiatric disorders. In all these cases, CSD was detected close to critical transitions. 

T. This has been observed in contexts such as blood pressure regulation, where slow recovery from changes in blood pressure has been linked to increased risk of mortality or of ischemic stroke [13–16] and in seizure prediction, where CSD has been reported in computational, in-vitro [17, 18], as well as in real data [19]. Critical slowing down is also thought to precede transitions in psychiatric disorders, including bipolar disorder and depression [20–22].

Q3. Mention any prior work that has looked at dynamics of vital signs around extubation, to position this study in the context of existing literature.

R. Despite the large number of studies on understanding and predicting extubation failure, investigations on the dynamics of vital signs near extubation failure have been fewer. We have now expanded the introduction with details on a few relevant studies which showed how dynamics of vital signs relate to extubation outcome. 

T. A smaller number of studies have investigated extubation failure by studying the dynamics of vital signs time series. White et. al. demonstrated the association between lower interbreath interval complexity and extubation failure[40]. Seely et. al. showed that the heart and respiratory rate variability recorded prior to extubation was significantly associated with extubation outcome [41]. Keim-Malpass et. al. investigated how adding variables based on the dynamics of vital signs resulted in a significant increase in the predictive capacity of a standard model based solely on static measures[42]. A number of recent studies have also attempted to predict extubation outcome using various recurrent neural networks architectures, with varying degrees of success [43-45]. It has also been shown previously that slow recovery to baseline levels of minute ventilation following successful spontaneous breathing trials is a risk factor for failure of extubation[46,47]. Such slow recovery to baseline levels aligns with what would be observed as a result of CSD prior to critical transitions in nonlinear dynamical systems [13].

Q Methods

Q4. The data collection and cohort segregation process could be explained in more detail in the main text, in addition to the new Figure 1. Specifically:Provide the total number of extubations analyzed and how many failed, to give a sense of the sample size.Clarify if any exclusion criteria were applied (e.g. minimum duration of data before/after extubation).

R. In addition to the figure 1, the number of analysed extubations and the corresponding failures have now been mentioned in the main text. These numbers correspond to numbers prior to pre-processing steps, hence excluding data losses due to the minimum duration of 2 hours required between extubation and reintubation (detailed from lines 175 in the manuscript). The final numbers after such data losses are presented in a new Table 2 in the manuscript, and detailed further in response to Q8.

T. As seen in Figure 1, cohorts 1, 2, 3 and 4 have sizes of 184, 2479, 333 and 38 respectively, prior to pre-processing. For patients who did not die in care, this corresponds to a failure rate of about 6.9 %.

Moreover, since the calculation of the quantifiers use windows of 60 minutes of data (see section 2.5), any ICU visit where the duration between the failed extubation and critical point (either re-intubation or death) is less than 120 minutes was removed. This is so that the window size is not greater than half of the length of the time series, which would fail to pick up nuances in the data and provide inaccurate estimations.

Q5. The statistical methods section could be expanded to:

Define the CSD metrics (autocorrelation, variance) more precisely and explain how they are calculated from the vital sign time series.

R. The CSD metrics are now explained with equations, detailing how the quantifiers are calculated for each window.

T. For a time series X = {x1, x2, ..., xn}, a window of size m and starting time t, would be given as X{t,m} = {xt, xt+1, ..., xt+m}. The variance and autocorrelation of each window would then be given by

(Eqns 3 and 4)

The resulting σ2,t and rt1 form a secondary time series, as the start point of the window,t, is slid forward, which shows how these measures change over time

Q6. Justify the choice of a 5% significance threshold for the CSD metric increases.

R. This exploration is the first that explores CSD in the context of extubation outcome. Therefore we could not estimate a significance threshold based on expected effect sizes. Hence the commonly used threshold of 5% (α = 0.05) was used for detecting increases in the CSD metric. Since multiple tests are being conducted, we also used a Holm-Bonferroni approach to correct for these.

T. Since the application of CSD to detect the need for reintubation is a novel approach, it is important to minimize the risk of missing clinically relevant changes. Hence the most commonly used 𝛼 level of 0.05 was chosen over a stricter threshold, while also correcting for multiple testing using the Holm-Bonferroni approach.

Q7. Describe the logistic regression model in more detail - what were the predictor variables and how were the magnitudes of CSD metric increases incorporated?

R. The full logistic regression models used are now listed in the main text, with details of the results including the log of the odds, standard error, z-values, p-values, degrees of freedom, AIC and R2 given in Table 6.

T. The logistic regression models used in the analysis are defined as follows.

(List of logistic regression models used)

Here Variabler1𝜏 and Variable𝜎2𝜏 represent the magnitude of the Kendall correlation coefficient for the autocorrelation and variance for each variable (HR,RR and ABP). The full results of the logistic regression models, including the logs of the odds (estimate), standard errors, z-values, p-values, degrees of freedom (N), Akaike information criterion (AIC) and R2 values are summarised in Table 6.

Q. Results

Q8. Report the total number of extubations analyzed and failure rate, to provide context 

for the findings.

R. The results section is now introduced with the final numbers of analysed files for each cohort and the corresponding extubation failure rate.

T. In the following sections, we present the results of the statistical analyses conducted ,including z-tests for significant increases in EWS, t-tests for comparing the magnitudes of increase, and a series of exploratory analyses to further investigate the trends in the data. The final number of files analyzed per ICU ward, cohort, and variable is shown in Table 2. This results in failure rates ranging from 5.4% to 6.4%, depending on the variable used without subdivision into wards, and from 3.9% to 14.8% when considering wards separately.

Q9. Providing the mean and standard deviation of the CSD metric increases for the success and failure groups, not just the p-values and effect sizes.

R. Table 4 is now modified to include the means and standard errors of the Kendall tau correlation coefficients for the success and failure groups, in addition to the p-values.

T. The means, standard errors and p-values corresponding to each of these tests are shown in Table 4 with the significant results highlighted.

Q10. Reporting the full logistic regression model results (coefficients, p-values, odds ratios) rather than just the model comparison.

R. The full results of the logistic regression model including the estimates (log of the odds ratio), standard errors, z-values, p-values, degrees of freedom, AIC and R2 values which were listed in Table 5 are highlighted further in the main text. In addition, the values for the intercept for each of the models which were previously missing have now been added to Table 5. In addition, some errors in the final estimates of the results in the previous version have been corrected.

T. The full results of the logistic regression models, including the logs of the odds (estimate), standard errors, z-values, p-values, degrees of freedom (N), Akaike information criterion (AIC) and R2 values are summarised in Table 6.

Q.Discussion

Q11. Acknowledge the limitations of the study, such as the small sample size and single-center nature of the data.

R. We thank the editor for pointing out these important limitations of the study. We have now highlighted these in the discussion section of the manuscript.

T. Although the small sampling time of the vital signs time series is rare in similar studies on extubation failure, the sample size of the data is comparatively small, especially in comparison to the MIMIC datasets which have been used extensively to study extubation failure [71,72]. Furthermore, we use only data from a single paediatric hospital in London, and our results may not directly translate to an adult population.

Q12. Discuss potential confounding factors that could influence the vital sign dynamics, such as medications, fluid status, etc.

R. Potential confounding factors such as medication and fluid status have now been mentioned in the discussion section, with appropriate references.

T. Moreover, a number of potential confounding factors could influence the dynamics of the vital signs considered in this work. These include influences from drug based interventions which could alter heart rate and blood pressure, as well as the patient's fluid balance which relates to respiratory and cardiovascular stability[66,67].

Q13. Speculate on how CSD metrics could potentially be used clinically in the future (e.g. as part of a predictive model for extubation failure) while acknowledging the current limitations.

R. The concluding lines have been expanded to include speculation of how CSD metrics could be used in clinical practice in future, while mentioning the limitations pointed out in our results.

T. While our results indicate that CSD metrics on their own may not be a reliable predictor of extubation outcome, they could constitute novel features in machine learning based prediction models and enhance their predictive power [42]. This would be a promising direction for future

research to potentially use CSD metrics in clinical practice, which would in turn be a significant stride towards using dynamical systems theory for prediction in the ICU.

R. We thank the editor again for the insightful comments on the manuscript. We hope that with these changes we have largely addressed their concerns and the manuscript can now be considered for publication in PLoS One.

---

## [Decision Letter · Decision Letter 2]

2 Oct 2024

PONE-D-24-07937R2Transitions in intensive care: Investigating critical slowing down post extubationPLOS ONE

Dear Dr. George,

Thank you for submitting your manuscript to PLOS ONE. After careful consideration, we feel that it has merit but does not fully meet PLOS ONE’s publication criteria as it currently stands. Therefore, we invite you to submit a revised version of the manuscript that addresses the points raised during the review process.

**Need revison for new points as asked by Reviewer with proper explaination.**

We look forward to receiving your revised manuscript.

Kind regards,

Lalit Gupta

Academic Editor

PLOS ONE

Reviewers' comments:

Reviewer's Responses to Questions

**Comments to the Author**

1. If the authors have adequately addressed your comments raised in a previous round of review and you feel that this manuscript is now acceptable for publication, you may indicate that here to bypass the “Comments to the Author” section, enter your conflict of interest statement in the “Confidential to Editor” section, and submit your "Accept" recommendation.

Reviewer #4: (No Response)

2. Is the manuscript technically sound, and do the data support the conclusions?

Reviewer #4: Yes

3. Has the statistical analysis been performed appropriately and rigorously? 

Reviewer #4: Yes

4. Have the authors made all data underlying the findings in their manuscript fully available?

Reviewer #4: Yes

5. Is the manuscript presented in an intelligible fashion and written in standard English?

Reviewer #4: Yes

6. Review Comments to the Author

Reviewer #4: This study concerns an investigation of the presence of critical slowing down (CSD) prior to failed and succesful extubation events in pediatric patients. The presence of CSD as signaled by rising autocorrelations and variances was examined in several physiological variables. Results showed very limited predictive promise, and the authors consequently conclude that this method is not clinically useful yet. On the whole, this investigation is well-written and despite the modest findings, it is a worthwhile addition to the growing literature on CSD-based early warning signals in medicine.

I was happy to see that the authors included cases in which CSD was not expected so that they can give a better understanding of the presence of EWS by calculating predictive values. Moreover, I appreciated the use of a single-case analysis to inform and preregister the methods of the current study.

After reading the manuscript, I have several comments that deserve to be addressed prior to publication. I will begin with the more major comments and list minor comments below.

Major comments:

1. Like many other studies on CSD in medicine to date, this study’s results do not yet provide much encouragement to pursue these methods in the context of clinical predictions. While combinations of variables and additions to machine learning models may yet improve the predictive abilities, much remains unknown about how to identify transition points, and how best to calculate CSD (relevant time scale, required data length and density, window size in moving-window analyses, AR/SD vs flickering, etc.). While the authors do note these difficulties in their discussion, I would like to see them make more concrete recommendations for future research.

1.1. For instance, the authors comment on several inherent limitations in the accurate categorization of true and false positives in the discussion (lines 420-437). Their relatively concrete outcome categorization is a strength in this study, even if it may still be somewhat subjective. I would like to see a more fleshed-out discussion of how these problems might be addressed in future studies. How do they envision that transition points can be more accurately identified? And in a real-time setting, at that.

1.2. Likewise, much remains unknown about the optimal way to analyze time series data to identify CSD. Do the authors have any suggestions on how to systematically examine the hurdles they list (and I listed above)?

1.3. Moreover, while their group-level combined logistic regression model shows somewhat better outcome prediction, this is not all that useful at an individual level. Can the authors translate this part of their study to avenues for future studies? For instance, do they recommend that studies combine multiple variables and signals to improve their predictive performance at single-case level? Or should the group-level results inform a different direction of study (e.g., machine-learning models only).

1.3.1. Have they considered using combinations of variables at single-case level for their own study?

2. I expected to see some discussion of the distribution of tau values they found. A stronger positive tau value over the autocorrelation and variance time series is taken to be a sign of CSD. However, in Figure 4 we also see that many negative trends in tau were found (in both C1 and C2). What might explain this? Are there theoretical or practical factors (data quality, time scale?) that could account for this?

I am aware of at least one more theoretical article that shows that declining trends in the variance may occur in particularly noise or insufficient data (Dakos, Van Nes et al., 2012 doi.org/10.1890/11-0889.1). The autocorrelation is expected to be more robust to this.

3. On line 336-338, the authors discuss the sensitivity and specificity values they found and suggest that in the few cases (~20%) that the model gives a warning signal, the ~80% specificity means that those cases might actually be mostly correct. I do not believe this is an entirely correct interpretation.

The specificity or true negative rate indicates the proportion of cases in which a true negative is correctly identified (thus, Cohort 2 is classified as not having an EWS), and does not directly relate to the correctness of positively flagged cases (Cohort 1). In fact, since ~20% of Cohort 2 is incorrectly flagged as a positive (the false negative rate, 1-Specificity) and the sample is so much larger, the chances of a positively flagged test actually being a true positive are incredibly small, around 7% (as seen in the PPV). To put it very clearly, the false discovery rate is nearly 94% (1-PPV) across their measures, so the interpretation they have given is not supported.

My recommendation to the authors is threefold:

3.1. Rewrite the interpretation on line 336-338 to more accurately reflect their findings, they are very modest at best.

3.2. Use a metric of the accuracy that accounts for unbalanced samples. For instance, the Balanced Accuracy ((Sensitivity + Specificity)/2) does not rely on the group sizes. The F1-score is another option, as it gives a higher weight to finding true positives, though it does still rely on group ratios (as per their own note about PPV/NPV on line 340).

3.2.1. Out of curiosity I calculated both those metrics from the confusion values in Table 5, and the Balanced Accuracy was around .5 (chance-level in a balanced dataset) across their measures, while the F1 scores were around .1. Both metrics range from 0 to 1, with 1 being optimal prediction, so a restrained conclusion about their findings is called for.

3.3. To more fairly represent their findings, and to tie in with the clinical nature of their data, a discussion of the number of false negatives (missed signals) is also warranted. Missing three quarters of the cases one would actually want to identify to prevent extubation failure, is obviously quite problematic. The authors could discuss how they might work on improving the sensitivity (changes to their analytical methods?), even if at the cost of specificity. Please elaborate further on the low specificity and missed signals in the discussion, alongside further interpretation of any new metrics that might be added in response to my previous point.

4. Overall, I recommend that the authors are more circumspect about their findings and how much promise they hold. While the discussion has not been written to give overly optimistic interpretations or inflated importance to their results, it can be stated more clearly that their findings are mostly null, with very poor predictive values. Perhaps the possibility that using CSD to predict transitions in medicine is NOT a worthwhile avenue to investigate further should be mentioned in light of their, and other studies’ limited findings. As mentioned above, and by the authors themselves, studying CSD in medicine is difficult, and centring their discussion more around the prerequisite advancements in theoretical and methodological understanding of biological systems might be fitting.

Minor notes:

On line 87, the term EWS is first introduced, but it is never written out in full. It may also be worth briefly explaining the relation between CSD and EWS for readers who are less embedded in this topic.

Page 6, Figure 1: the flowchart includes the same outcome twice “Monitor data between extubation and pseudo-reintubation times”. Is that correct? The note mentions for Cohort 1 and 3 “monitor data … between extubation and re-intubation” – so without pseudo. Please make sure the figure is correct.

Typo on line 212: the word ‘variance’ here should be ‘autocorrelation’, the topic of the hypothesis.

For Table 3, is the title as intended? It currently reads “The proportions of significant Mann-Kendall hypothesis tests conducted on cohort 1” but it reports both C1 and C2 and the different variables and EWS.

Line 283: the authors write “a greater value of tau”, the authors could consider specifying that a greater positive value of tau is what is expected. It may be clear from the tested hypotheses that follow shortly after, but it cannot hurt to use precise language here. That would also help set up a later discussion of their results, as I have suggested above in comment 2.

Table 5, similar issue to Table 5 in that the table is described as referring to Cohort 1 only (and includes this label in the first column header). However, the authors are reporting true and false positives and negatives. The FP and TN values are based on cohort 2, so I recommend that the authors update their labels.

7. PLOS authors have the option to publish the peer review history of their article (what does this mean?). If published, this will include your full peer review and any attached files.

Reviewer #4: No

---

## [Author Response · Author response to Decision Letter 2]

5 Dec 2024

PONE-D-24-07937R2: Response to Referees

We thank the referee for their questions and comments. Modifications made in response to these have improved the manuscript considerably, and we hope that the referee is satisfied with the revisions made. Queries are marked with a Q, and our responses are provided with an R. Changes to the text are marked with a T.

Reviewer #4: 

Q. This study concerns an investigation of the presence of critical slowing down (CSD) prior to failed and successful extubation events in pediatric patients. The presence of CSD as signaled by rising autocorrelations and variances was examined in several physiological variables. Results showed very limited predictive promise, and the authors consequently conclude that this method is not clinically useful yet. On the whole, this investigation is well-written and despite the modest findings, it is a worthwhile addition to the growing literature on CSD-based early warning signals in medicine.

I was happy to see that the authors included cases in which CSD was not expected so that they can give a better understanding of the presence of EWS by calculating predictive values. Moreover, I appreciated the use of a single-case analysis to inform and preregister the methods of the current study.

R. We thank the reviewer for their accurate summary and their comments on the findings. As the reviewer points out, our work was pre-registered in an effort to limit biases that could arise as a result of extensive explorations in the dataset. Although the original single-case analysis showed promise, the final analysis on the entire sample indicated that CSD is rare prior to re-intubation.

Q. After reading the manuscript, I have several comments that deserve to be addressed prior to publication. I will begin with the more major comments and list minor comments below.

R. In line with the reviewers recommendations, we have now significantly changed the results and discussion section to address many of the points raised. New subsections have been added to the discussion section to improve clarity and accessibility for readers. Many of the answers and changes made to the manuscript feed into multiple questions raised by the reviewer. For instance, improving accurate identification of transition points may lead to higher performance in future studies. In such instances the answers are only mentioned in response to the questions where they are most relevant for the sake of brevity. Changes to the manuscript are mentioned in response to the queries in red, with grey lines indicating sections from the previous version of the manuscript which are added for clarity.

Q. Major comments:

1. Like many other studies on CSD in medicine to date, this study’s results do not yet provide much encouragement to pursue these methods in the context of clinical predictions. While combinations of variables and additions to machine learning models may yet improve the predictive abilities, much remains unknown about how to identify transition points, and how best to calculate CSD (relevant time scale, required data length and density, window size in moving-window analyses, AR/SD vs flickering, etc.). While the authors do note these difficulties in their discussion, I would like to see them make more concrete recommendations for future research.

R. We thank the reviewer for this comment. As the reviewer rightly points out, this study aligns with many other investigations on CSD in medicine, most of which show limited potential for clinical prediction. The reviewer points out a number of difficulties in identifying transition points and calculating CSD in data. Each of these points are indeed relevant to the discussion and have now been added in the new version of the manuscript. Subsections have now been added, expanding on the previous version of the manuscript, on 4.3 Reasons for low model performance, 4.4 Limitations and 4.5 Future directions. Specifics are discussed in response to the sub points raised below

Q 1.1. For instance, the authors comment on several inherent limitations in the accurate categorization of true and false positives in the discussion (lines 420-437). Their relatively concrete outcome categorization is a strength in this study, even if it may still be somewhat subjective. I would like to see a more fleshed-out discussion of how these problems might be addressed in future studies. How do they envision that transition points can be more accurately identified? And in a real-time setting, at that.

R. We thank the reviewer for this question. The categorization of extubation failures and determination of exact extubation times is indeed a strength of this study. Despite this, a number of systematic issues remain in the objective identification of transition points in this study. Some of these issues were pointed out in the previous version of the discussion. New additions to the discussion section are now shown below.

T. Section 4.4 Limitations

As mentioned above, a major limitation in studies on extubation is in identifying true physiological transition points. In clinical practice, the decision to re-intubate is made preemptively in anticipation of potential instability, rather than waiting for definitive evidence of a transition. As a result transition points in this study may not indicate a change in the dynamics of the system, unlike in other fields where EWS have been detected. This study uses the timing of re-intubation as an approximate transition point, which may often be far from the true transition point.

Section 4.5 Future directions

To overcome the limitation of the subjectivity associated with transition points, future studies could focus on instances where emergency reintubation was required. In such cases, the actual point of reintubation may be much closer to the transition point than in a general sample. Another possibility could be to quantify markers of instability, such as deviations in physiological baselines using methods such as change point analysis. Only re-intubations that also include significant quantitative change would then be checked for early warning signals, as done in studies on CSD in other clinical contexts [85].

Q. 1.2. Likewise, much remains unknown about the optimal way to analyze time series data to identify CSD. Do the authors have any suggestions on how to systematically examine the hurdles they list (and I listed above)?

R. Many methodological papers have considered how to analyse time series data for identifying CSD either in general or specific contexts. Based on the literature and our findings, general guidelines for detrending, accounting for serial correlations, and choosing window sizes may be effective but require context-specific adjustments. Many of the choices in this paper were made in line with literature and tested out in a single case. The null results of this study suggest that the issues identified in the single individual may not generalize to the whole dataset. Perhaps future studies would benefit from using a small subsample instead of a single dataset to determine the best parameters for detecting CSD in the entire data. Additional considerations for analysing time series data to identify CSD have been listed in section 4.3 and mentioned in response to point 3.3.

T. 4. Discussion

4.3 Reasons for low model performance

In our study we attempted to overcome these difficulties by systematically examining and overcoming them in a single test case [48]. However the largely negative results of our study on the entire dataset seems to suggest that either the issues identified on the single dataset did not generalize to the whole system. To avoid this, future studies in such complex biological datasets may benefit from using a small, representative subset (perhaps 5%) of the dataset to optimize parameters for identifying CSD.

Q. 1.3. Moreover, while their group-level combined logistic regression model shows somewhat better outcome prediction, this is not all that useful at an individual level. Can the authors translate this part of their study to avenues for future studies? For instance, do they recommend that studies combine multiple variables and signals to improve their predictive performance at single-case level? Or should the group-level results inform a different direction of study (e.g., machine-learning models only).

R. The reviewer correctly points out that the results of the logistic regression model is a group level analysis which does not directly translate to the level of the individual. One way to translate this analysis to an individual level is to calculate the mean and CSD variables over a sliding window and calculate a combined metric that predicts extubation outcome. And as already mentioned in the previous iteration of the manuscript, CSD metrics could also be used as a feature to train machine learning models, which could then be used to predict need for reintubation in new unseen samples. These have now been added in the recommendations for future research.

T. 4. Discussion

4.5 Future Directions

Adding CSD metrics to the mean levels of the variables also showed a significant improvement over the model without them. At an individual level this could indicate that the mean and CSD metrics calculated over a sliding window could predict extubation outcome better than using either of them alone. Going even further, while CSD metrics on their own may not be a reliable predictor of extubation outcome, they could constitute novel features in machine learning based prediction models and enhance their predictive power[42].

Q. 1.3.1. Have they considered using combinations of variables at single-case level for their own study?

R. While this was not considered before, we have now done two additional analyses at the group level to check whether this would yield better results. The first tested how the sum of strengths of positive transitions (measured as the sum of positive 𝜏 values) differed between Cohorts 1 and 2. We saw that Cohort 1 had a significantly higher value for this than Cohort 2 in a one sided t-test. The AUC calculated by varying this metric gave a value of 0.597, higher than any of the individual metrics. We have now added these to the results section and discussed the implications for future studies in the subsection on future directions. 

T. 3. Results

3.4 Model performance

Since model 4 that combines the HR,RR and ABP variables performed better than models 1,2 and 3 individually, a one-sided t-test was conducted to check if Cohort 1 had a higher strength of positive transitions across variables (defined as the sum of positive 𝜏 or all variables) as compared to Cohort 2, yielding significant results (t-statistic=2.056, p-value=0.022). Moreover, the AUC of the combined strength of positive transitions was 0.597, higher than the AUC for the individual variables.

4. Discussion

4.4 Future Directions

A significant outcome of the present study is the improvement seen by using CSD variables in a simple logistic regression model. At the group level, the combination of CSD variables showed lower AIC and higher R2 than models using individual CSD variables. This difference persisted even when the degrees of freedom were kept the same. This indicated that a combination of variables could perform better than individual CSD variables to detect extubations. Moreover Cohort 1 showed a higher strength of positive transitions across variables as compared to Cohort 2, and a higher predictive capacity (measured using AUC) as compared to the individual variables. This indicates that using a combination of EWS at an individual level could reduce misdetection and increase sensitivity in the sample.

Q. 2. I expected to see some discussion of the distribution of tau values they found. A stronger positive tau value over the autocorrelation and variance time series is taken to be a sign of CSD. However, in Figure 4 we also see that many negative trends in tau were found (in both C1 and C2). What might explain this? Are there theoretical or practical factors (data quality, time scale?) that could account for this?

I am aware of at least one more theoretical article that shows that declining trends in the variance may occur in particularly noise or insufficient data (Dakos, Van Nes et al., 2012 doi.org/10.1890/11-0889.1). The autocorrelation is expected to be more robust to this.

R. The negative trends observed in Figure 4 are indeed quite interesting. Several factors, including noise or insufficient data, as noted by the reviewer, could contribute to such trends. Theoretical causes could include flickering prior to a transition or processes that exhibit critical speeding up. In complex biological data such as ours a combination of factors could be at play leading to the large number of negative trends observed in data. We have now included a paragraph discussing the distributions of tau values in Figure 4.

T. 4. Discussion

4.1 Summary and significance

One of the interesting features of the study is the distributions of the Kendall τ coefficients in Figure 4. Apart from strong false positives and negatives, we also see a number of negative trends in both cohorts 1 and 2. There is some evidence that such negative trends could occur in autocorrelation in systems exhibiting flickering prior to transitions or in both autocorrelation and variance as a consequence of processes that exhibit critical speeding up [65 –67]. The variance is also known to decrease when there is insufficient data and the fluctuations in the system are dominated by low frequencies or when the data is highly noisy [68]. In complex biological data, such as this, both issues related to data, as well as competing processes with opposite effects could lead to such negative trends.

Q. 3. On line 336-338, the authors discuss the sensitivity and specificity values they found and suggest that in the few cases (~20%) that the model gives a warning signal, the ~80% specificity means that those cases might actually be mostly correct. I do not believe this is an entirely correct interpretation.

The specificity or true negative rate indicates the proportion of cases in which a true negative is correctly identified (thus, Cohort 2 is classified as not having an EWS), and does not directly relate to the correctness of positively flagged cases (Cohort 1). In fact, since ~20% of Cohort 2 is incorrectly flagged as a positive (the false negative rate, 1-Specificity) and the sample is so much larger, the chances of a positively flagged test actually being a true positive are incredibly small, around 7% (as seen in the PPV). To put it very clearly, the false discovery rate is nearly 94% (1-PPV) across their measures, so the interpretation they have given is not supported.

My recommendation to the authors is threefold:

3.1. Rewrite the interpretation on line 336-338 to more accurately reflect their findings, they are very modest at best.

R. We thank the reviewer for pointing out this error. As they rightly point out, the inflated specificity is indeed simply due to the class imbalance in the problem, and not suggestive of the correctness of the positive class. The PPV correctly shows this, and the previous incorrect interpretation is now removed. The previous sentence interpreting the high specificity as the proportion of correctly identified cases is now removed and replaced with the following.

T. 3. Results

3.4 Model performance

The low sensitivity and PPV values indicate that the model has limited predictive power... Since a significant proportion of the much larger Cohort 2 is incorrectly flagged as a positive, the chances of a positively flagged test actually being a true positive are incredibly small, as seen in the PPV.

Q. 3.2. Use a metric of the accuracy that accounts for unbalanced samples. For instance, the Balanced Accuracy ((Sensitivity + Specificity)/2) does not rely on the group sizes. The F1-score is another option, as it gives a higher weight to finding true positives, though it does still rely on group ratios (as per their own note about PPV/NPV on line 340).

3.2.1. Out of curiosity I calculated both those metrics from the confusion values in Table 5, and the Balanced Accuracy was around .5

---

## [Decision Letter · Decision Letter 3]

23 Dec 2024

Transitions in intensive care: Investigating critical slowing down post extubation

PONE-D-24-07937R3

Dear Dr. George,

We’re pleased to inform you that your manuscript has been judged scientifically suitable for publication and will be formally accepted for publication once it meets all outstanding technical requirements.

Kind regards,

Lalit Gupta

Academic Editor

PLOS ONE

Additional Editor Comments (optional):

the manuscript is well-addressed to all the comments previously asked

Reviewers' comments:

Reviewer's Responses to Questions

**Comments to the Author**

1. If the authors have adequately addressed your comments raised in a previous round of review and you feel that this manuscript is now acceptable for publication, you may indicate that here to bypass the “Comments to the Author” section, enter your conflict of interest statement in the “Confidential to Editor” section, and submit your "Accept" recommendation.

Reviewer #5: All comments have been addressed

Reviewer #6: All comments have been addressed

2. Is the manuscript technically sound, and do the data support the conclusions?

Reviewer #5: Yes

Reviewer #6: Yes

3. Has the statistical analysis been performed appropriately and rigorously? 

Reviewer #5: (No Response)

Reviewer #6: Yes

4. Have the authors made all data underlying the findings in their manuscript fully available?

Reviewer #5: Yes

Reviewer #6: Yes

5. Is the manuscript presented in an intelligible fashion and written in standard English?

Reviewer #5: Yes

Reviewer #6: Yes

6. Review Comments to the Author

Reviewer #5: The contribution of this revised manuscript is clear and good. The author’s answer was very good and satisfied. Therefore, my recommendation is to accept the revised manuscript that has ref. no. PONE-D-24-07937R3 for publication.

for publication.

Reviewer #6: The authors present a well-structured study titled "Transitions in intensive care: Investigating critical slowing down post extubation" that explores the presence of critical slowing down (CSD) as an early warning signal preceding extubation failure in pediatric intensive care patients. This study addresses an important clinical problem and employs dynamical systems theory to examine physiological time series data (heart rate, respiratory rate, and mean blood pressure). Overall, the manuscript is well-addressed to all the comments.

7. PLOS authors have the option to publish the peer review history of their article (what does this mean?). If published, this will include your full peer review and any attached files.

Reviewer #5: No

Reviewer #6: No

---

## [Editor Report · Acceptance letter]

11 Jan 2025

PONE-D-24-07937R3 

PLOS ONE

Dear Dr. George, 

I'm pleased to inform you that your manuscript has been deemed suitable for publication in PLOS ONE. Congratulations! Your manuscript is now being handed over to our production team.

Kind regards, 

on behalf of

Dr. Lalit Gupta 

Academic Editor

PLOS ONE